# Epidemiological inference for emerging viruses using segregating sites

Yeongseon Park [1], Michael A. Martin [1,4] & Katia Koelle [2,3] ✉

Epidemiological models are commonly fit to case and pathogen sequence data to estimate parameters and to infer unobserved disease dynamics. Here, we present an inference approach based on sequence data that is well suited for model fitting early on during the expansion of a viral lineage. Our approach relies on a trajectory of segregating sites to infer epidemiological parameters within a Sequential Monte Carlo framework. Using simulated data, we first show that our approach accurately recovers key epidemiological quantities under a single-introduction scenario. We then apply our approach to SARS-CoV-2 sequence data from France, estimating a basic reproduction number of approximately 2.3-2.7 under an epidemiological model that allows for multiple introductions. Our approach presented here indicates that inference approaches that rely on simple population genetic summary statistics can be informative of epidemiological parameters and can be used for reconstructing infectious disease dynamics during the early expansion of a viral lineage.

Phylodynamic inference methods use pathogen sequence data to estimate epidemiological quantities such as the basic reproduction number and to reconstruct epidemiological patterns of incidence and prevalence. These inference methods have been applied to sequence data across a broad range of RNA viruses, including HIV[1–4], Ebola virus[5–7], dengue viruses[8], influenza viruses[9], and most recently severe acute respiratory syndrome coronavirus 2 (SARS-CoV-2)[10–12]. Most commonly, phylodynamic inference methods rely on underlying coalescent models or birth-death models. Coalescent-based approaches have been generalized to accommodate time-varying population sizes and structured epidemiological models, for example, susceptible-exposed-infected-recovered (SEIR) models and models with spatial subdivision[6,13]. Birth-death approaches[14,15], where a birth in the context of infectious diseases corresponds to a new infection and death corresponds to a recovery from infection, carry advantages such as capturing the role of demographic stochasticity in disease dynamics, which may be particularly important in emerging diseases that start with low infection numbers[16]. Birth-death approaches have also been expanded to incorporate the complex nature of infectious disease dynamics including structured populations[17]. Both coalescent-based and birth-death phylodynamic inference approaches rely on time-

resolved phylogenies and have been incorporated into the phylogenetics software packages BEAST1[18] and BEAST2[19] to allow for joint estimation of epidemiological parameters and dynamics while integrating over phylogenetic uncertainty[6,20]. Integrating over phylogenetic uncertainty is crucial when applying these methods to viral sequence data that are sampled over a short period of time and contain only low levels of genetic diversity. However, integrating over phylogenetic uncertainty can be computationally intensive. Moreover, phylodynamic approaches that use reconstructed trees for inference require estimation of parameters associated with models of sequence evolution, along with parameters that are of more immediate epidemiological interest.

Here, we present an alternative sequence-based statistical inference method that may be particularly useful when viral sequences are sampled over short time periods and when phylogenetic uncertainty present in time-resolved viral phylogenies is considerable. Instead of relying on viral phylogenies to infer epidemiological parameters or to reconstruct patterns of viral spread, the "tree-free" method we propose here fits epidemiological models to time series of the number of segregating sites (that is, the number of polymorphic sites) present in a sampled viral population. The approach we propose here allows for

[1]Graduate Program in Population Biology, Ecology, and Evolution, Emory University, Atlanta, GA 30322, USA. [2]Department of Biology, Emory University, Atlanta, GA 30322, USA. [3]Emory Center of Excellence for Influenza Research and Response (CEIRR), Atlanta, GA, USA. [4]Present address: Department of Pathology, Johns Hopkins University School of Medicine, Baltimore, MD, USA. ✉e-mail: katia.koelle@emory.edu

structured infectious disease models to be considered in a straight-forward "plug-and-play" manner. It also incorporates the effect that demographic noise has on epidemiological dynamics. Below, we first describe how segregating site trajectories are calculated using sequence data and how they are impacted by sampling effort, rates of viral spread, and transmission heterogeneity. We then describe our proposed statistical inference method and apply it to simulated data to demonstrate the ability of this method to infer epidemiological parameters and to reconstruct unobserved epidemiological dynamics. Finally, we apply our segregating sites method to SARS-CoV-2 sequence data from France, arriving at quantitatively similar parameter estimates to those arrived at using epidemiological data.

## Results

### Segregating site trajectories are informative of epidemiological dynamics

The number of segregating sites present in a set of sampled viral sequences is defined as the number of nucleotide sites at which genetic variation is present in the sample set. To determine whether the number of segregating sites that are observed over time in a viral population may be informative of underlying epidemiological dynamics, we forward-simulated a classic susceptible-exposed-infected-recovered (SEIR) epidemiological model, augmented with viral evolution, under various sampling efforts and parameterizations (Fig. 1; Methods). Simulations of this augmented SEIR model, initialized with a single infected individual, first indicate that segregating site trajectories are sensitive to sampling effort, as expected (Fig. 1a, b). More specifically, we considered three different sampling strategies, each with sequences binned in consecutive, nonoverlapping 4-day time windows to calculate segregating site trajectories. These three sampling strategies consisted of a strategy with full sampling effort (all sequences per 4-day time window), one with dense sampling effort

(40 sequences per 4-day time window) and one with sparse sampling effort (20 sequences per 4-day time window). With all three of these sampling efforts, the number of segregating sites first increases as the epidemic grows, with mutations accumulating in the virus population. Following the peak of the epidemic, the number of segregating sites starts to decline as viral sublineages die out, reducing the amount of genetic variation present in the viral population. A comparison between full, dense, and sparse sampling efforts indicates that lowering sampling effort results in a lower number of observed segregating sites during any time window. This is because at lower sampling effort, less of the genetic variation present in a viral population over a given time window is likely to be sampled. The patterns shown here across sampling strategies are robust to the time window length used for the calculation of segregating site trajectories (Figure S1).

To assess whether segregating site trajectories could be used for statistical inference, we first considered whether these trajectories differed between epidemics governed by different basic reproduction numbers ($R_0$ values). Figure 1c shows simulations of the SEIR model under two parameterizations of the basic reproduction number: an $R_0$ of 1.6, corresponding to the simulation shown in Fig. 1a, and a higher $R_0$ of 2.0 (implemented via a higher transmission rate $\beta$). The epidemic with the higher $R_0$ expanded more rapidly (Fig. 1c) and, under the same sampling effort, resulted in a more rapid increase in the number of segregating sites (Fig. 1d). This indicates that segregating site trajectories can be informative of $R_0$ early on in an epidemic.

We next considered the effect of transmission heterogeneity on segregating site trajectories. Many viral pathogens are characterized by 'superspreading' dynamics, where a relatively small proportion of infected individuals are responsible for a large proportion of secondary infections[21]. The extent of transmission heterogeneity is often gauged relative to the 20/80 rule (where the most infectious 20% of infected individuals are responsible for 80% of the secondary cases[22]).

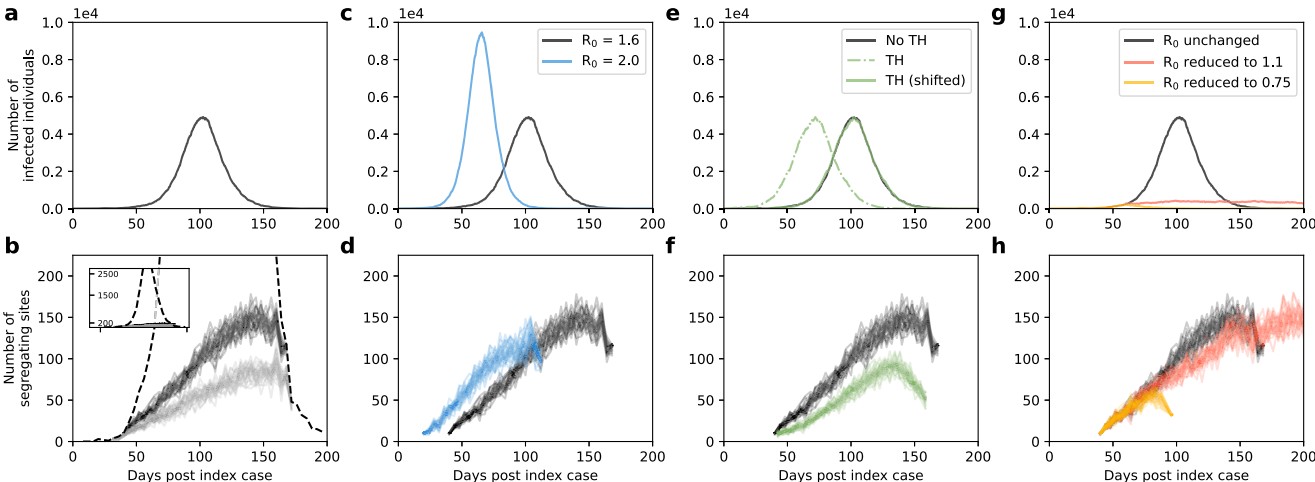

**Fig. 1 | Segregating site trajectories under simulated epidemiological dynamics. a** Dynamics of infected individuals ($I$) under an SEIR model simulated with an $R_0$ of 1.6. **b** Segregating site trajectories under full (black dashed line), dense (black lines), and sparse (gray lines) sampling efforts. Dense and sparse sampling correspond to 40 and 20 sequences sampled per time window, respectively. **c** Simulated infected dynamics under the SEIR model with an $R_0$ of 2.0 (blue line) compared to those of the $R_0 = 1.6$ simulation (black line). **d** Segregating site trajectories for the two simulations shown in panel **c**. **e** Simulated infected dynamics under the SEIR model with transmission heterogeneity (green, dashed line) compared to those of the $R_0 = 1.6$ simulation (black line) without transmission heterogeneity. Transmission heterogeneity was included by setting the parameter $p_h$ to 0.06. For ease of comparing segregating site trajectories, the transmission heterogeneity simulation was shifted later in time (green, solid line). **f** Segregating site trajectories for the shifted transmission heterogeneity simulation (green lines) and the original

simulation (black lines). **g** Simulated infected dynamics under the SEIR model with changing $R_0$. In the simulations shown in red and yellow, when the number of infected individuals reached 400, $R_0$ was decreased to 1.1 and 0.75, respectively. The simulation in black has $R_0$ remaining at 1.6. **h** Segregating site trajectories for the three simulations shown in panel **g**. Dense sampling effort was used to generate all segregating site trajectories shown in panels **d**, **f**, and **h**. 30 randomly-sampled segregating site trajectories are shown for each sampling effort in panel **b** and for each epidemiological scenario in panels **d**, **f**, and **h**. In all model simulations, $\gamma_E = 1/2$ days⁻¹, $\gamma_I = 1/3$ days⁻¹, population size $N = 10^5$, and the per genome, per transmission mutation rate $\mu = 0.2$. Initial conditions are $S(t_0) = N-1$, $E(t_0) = 0$, $I(t_0) = 1$, and $R(t_0) = 0$. For the transmission heterogeneity simulation (panel **e**), $I_h(t_0) = 1$ and $I_l(t_0) = 0$ was used instead of $I(t_0) = 1$. A time step of $\tau = 0.1$ days was used in the Gillespie $\tau$-leap algorithm.

Some pathogens like SARS-CoV-2 exhibit extreme levels of super-spreading, with as low as 10-15% of infected individuals responsible for 80% of secondary cases[10,23–25]. Because transmission heterogeneity is known to impact patterns of viral genetic diversity[26], we simulated the above SEIR model with transmission heterogeneity to ascertain its effects on segregating site trajectories (Methods). Because transmission heterogeneity has a negligible impact on epidemiological dynamics once the number of infected individuals is large[27], epidemiological dynamics with and without transmission heterogeneity should be quantitatively similar to one another, with transmission heterogeneity simply expected to shorten the timing of epidemic onset in simulations with successful invasion[21]. Our simulations, parameterized with extreme transmission heterogeneity of 6/80, confirm this pattern (Fig. 1e). To compare segregating site trajectories between these simulations, we therefore shifted the simulation with transmission heterogeneity later in time such that the two simulated epidemics peaked at similar times (Fig. 1e). Comparisons of segregating site trajectories between these simulations indicated that transmission heterogeneity decreased the number of segregating sites during every time window (Fig. 1f). As expected, lower levels of transmission heterogeneity result in less substantial decreases in the number of segregating sites (Figure S2). Together, these results indicate that transmission heterogeneity needs to be taken into consideration when estimating epidemiological parameters using segregating site trajectories.

Finally, we wanted to assess whether changes in $R_0$ over the course of an epidemic would leave signatures in segregating site trajectories. We considered this scenario because phylodynamic inference has often been used to quantify the effect of public health interventions on $R_0$, most recently in the context of SARS-CoV-2[10,11]. We thus implemented simulations with $R_0$ starting at 1.6 and then either remaining at 1.6 or reduced to either 1.1 or 0.75 when the number of infected individuals reached 400 (Fig. 1g). The segregating site trajectories for these three simulations indicate that reductions in $R_0$ over the course of an epidemic leave signatures in this summary statistic of viral diversity (Fig. 1h). The signatures left in the segregating site trajectories reflect the epidemiological dynamics that result from the reductions in $R_0$. Reducing $R_0$ to 1.1 results in a slower increase in the number of cases and a delayed, as well as broader, epidemic peak; as such, the number of segregating sites increases more slowly and the decline in the number of segregating sites is not apparent over the time period shown. Reducing $R_0$ to 0.75 results in an immediate decline in cases, with an observed drop in the number of segregating sites due to the stochastic loss of viral sublineages. Similar magnitude reductions in $R_0$ that were implemented later on in the simulated epidemic yielded fainter signatures of this effect in the segregating site trajectories (Figure S3).

## Epidemiological inference using segregating site trajectories

To examine the extent to which inference based on segregating sites can be used for epidemiological parameter estimation, we generated a mock segregating site trajectory by forward simulating an SEIR model with an $R_0$ of 1.6. From this simulation, we randomly sampled 500 viral sequences (corresponding to approximately 0.78% of infections being sampled) and binned these sequences into 4-day time windows based on their sampling times (Fig. 2a). Figure 2b shows the segregating site trajectory from these binned sequences. From this trajectory, we first attempted to estimate only $R_0$ under the assumption that the timing of the index case $t_0$ is known (Methods). We estimated an $R_0$ value of 1.58 (95% confidence interval of 1.37 to 1.81; Fig. 2c), demonstrating that our segregating sites inference approach applied to this simulated dataset is able to recover the true $R_0$ value of 1.6. Lower levels of sampling effort (100 viral sequences) resulted in an $R_0$ estimate to 1.65 and a broader 95% confidence interval (1.30 to 2.06; Figure S4). Instead of random sampling of sequences, adopting a more uniformly distributed sampling strategy acted to reduce the uncertainty in the $R_0$ estimate (Figure S5). In Figure S6, we present results for the same set of sequences as those used in Fig. 2, with the sequence data binned instead in time windows of 1 day, 2 days, 6 days, and 10 days, rather than in a time window of 4 days. These results show that $R_0$ estimates are not biased by the use of different time window lengths.

Because the timing of the index case $t_0$ (in cases with a single introduction) is almost certainly not known for an emerging epidemic, we further attempted to estimate both $R_0$ and $t_0$ using the segregating site trajectory shown in Fig. 2b. We considered a range of $R_0$ values between 1.0 to 2.5 and a broad range of $t_0$ starting 50 days prior to the true start date of 0 and ending at the date of the first sampled sequence. We divided this parameter space into fine-resolution parameter combinations ($R_0$ intervals of 0.1 and $t_0$ intervals of 2 days) and ran 20 SMC simulations for every parameter combination. In Fig. 3a, we plot the mean value of the 20 SMC log-likelihoods for every parameter combination in the considered parameter space. Examination of this plot indicates that there is a log-likelihood ridge that runs between early $t_0$/low $R_0$ parameter sides, indicating that inference using

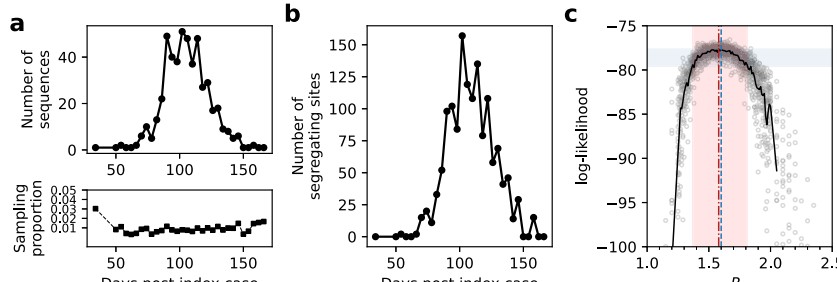

**Fig. 2 | Epidemiological inference on a simulated trajectory of segregating sites.** **a**, top The number of sampled sequences over time, binned by 4-day time windows. Sampling was done in proportion to the number of individuals recovering in a time window. In all, 500 sequences were sampled over the course of the simulated epidemic. **a**, bottom The proportion of sampled individuals in each time window, obtained by dividing the number of sampled individuals by the number of individuals who recovered during a time window. **b** Simulated segregating site trajectory from the sampled sequences, by time window. **c** Estimation of $R_0$ using Sequential Monte Carlo (SMC). Points show log-likelihood values from different SMC simulations. $R_0$ values between 1.0 and 1.25 and between 2.0 and 2.5 were considered with a step size of 0.1. $R_0$ values between 1.25 and 2.0 were considered with a step size of 0.01. Solid black curve shows the mean of 20 data points for each $R_0$ value. The vertical red dashed line shows the maximum likelihood estimate (MLE) of $R_0$. The red band shows the 95% confidence interval of $R_0$. The vertical blue line shows the true value of $R_0 = 1.6$. The MLE and 95% CI were obtained using the mean log-likelihood values. The 95% CI band included the set of $R_0$ values with log-likelihoods that fell within 1.92 units of the highest mean log-likelihood value, based on a chi-squared distribution with 1 degree of freedom. Model parameters for the simulated data set are: $R_0 = 1.6$, $\gamma_E = 1/2$ days⁻¹, $\gamma_I = 1/3$ days⁻¹, population size $N = 10^5$, $t_0 = 0$, and the per genome, per transmission mutation rate $\mu = 0.2$. Initial conditions are $S(t_0) = N-1$, $E(t_0) = 0$, $I(t_0) = 1$, and $R(t_0) = 0$. A time step of $\tau = 0.1$ days was used in the Gillespie $\tau$-leap algorithm.

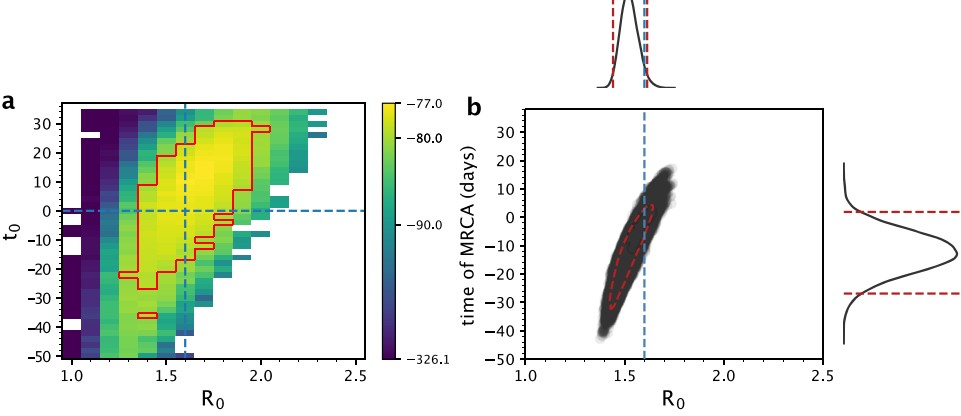

**Fig. 3 | Joint estimation of the basic reproduction number ($R_0$) and the timing of the index case ($t_0$) using simulated data, and comparison against PhyDyn.** **a** The log-likelihood surface based on the segregating site trajectory shown in Fig. 2b is shown over a range of $R_0$ and $t_0$ parameter combinations. The log-likelihood value shown in each cell is the mean log-likelihood value calculated from 20 SMC simulations. Blank cells yielded mean log-likelihood values of negative infinity. The red boundary shows the set of ($R_0$, $t_0$) values that fall within the 95%

confidence region. Parameter combinations within the red boundary have mean log-likelihood values that fall within 2.996 units of the highest mean log-likelihood value, based on a chi-squared distribution with 2 degrees of freedom. **b** Joint density plot for $R_0$ and the time of the most recent common ancestor (tMRCA), as estimated using PhyDyn[6] on the same set of 500 sampled sequences. Dashed red line in the joint density plot shows the 95% HPD interval of the joint density.

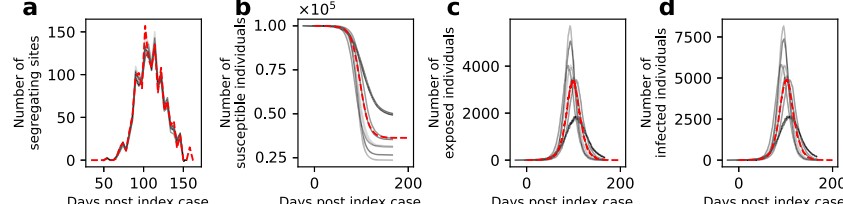

**Fig. 4 | Reconstruction of unobserved state variables.** **a** Simulated trajectory of the number of segregating sites (dashed red), alongside reconstructed trajectories of the number of segregating sites (gray). **b** Simulated dynamics of susceptible individuals (dashed red), alongside reconstructed dynamics of susceptible individuals (gray). **c** Simulated dynamics of exposed individuals (dashed red), alongside reconstructed dynamics of exposed individuals (gray). **d** Simulated dynamics of infected individuals (dashed red), alongside reconstructed dynamics of infected individuals (gray). Reconstructed state variables were obtained by running the

particle filter using $R_0$ and $t_0$ parameter values randomly sampled from within the 95% CI region, with a further condition that the log-likelihood from the run exceeded the 95% CI region log-likelihood cutoff shown in Fig. 3a. To show that resampling of particles during the SMC performs effectively, we show in Figure S7 the dynamics of these unobserved state variables in particles that are sampled at different time points during the SMC procedure that may be lost by the end of the simulation as a result of resampling.

segregating site trajectories can in principle estimate both $t_0$ and $R_0$. The parameter combination with the highest mean log-likelihood was $R_0 = 1.7$ and $t_0 = 16$ days, with the true parameter combination of $R_0 = 1.6$ and $t_0 = 0$ days falling within the 95% confidence region of the estimated parameters. Our results therefore indicate that joint estimation of these parameters is thus possible in cases where a single introduction is responsible for igniting local circulation. Using our estimates of $R_0$ and $t_0$, we reconstructed the dynamics of the segregating sites (Fig. 4a) and unobserved state variables: the number of susceptible, exposed, and infected individuals over time (Fig. 4b-d). These reconstructed state variables captured the true epidemiological dynamics, demonstrating that our segregating sites approach can be used to infer epidemiological variables that generally go unobserved.

As mentioned in the Introduction, there are existing phylodynamic inference approaches available that can estimate epidemiological model parameters using viral phylogenies that have been reconstructed from sequence data. Of particular note is the coalescent-based inference approach developed by Volz[13] that has been implemented as PhyDyn[6] in BEAST2. To compare our results using the segregating sites approach to results using PhyDyn, we generated mock viral nucleotide sequences from our set of 500 sampled sequences (Methods) and used these nucleotide sequences as input into PhyDyn. Assuming the same epidemiological model structure and using uninformative priors, PhyDyn was similarly able to

recover the true $R_0$ value of 1.6 used in the forward simulation (Fig. 3b; 95% credible interval = 1.44 to 1.61). Because PhyDyn infers epidemiological parameters using a tree-based method, the program does not estimate the time of the index case $t_0$. Instead, it estimates the time of the most recent common ancestor (tMRCA) of the viral phylogeny. The credible interval of PhyDyn's tMRCA estimate spanned from −26.89 to 1.87 days post the true time of the index case ($t_0 = 0$). Times of a most recent common ancestor, however, are generally later (and never earlier) than the time of the index case. This is because some viral lineages likely go unsampled and the pruning of these unsampled lineages results in a tMRCA that can be considerably later than the time of the index case $t_0$[28]. As such, interpretation of the PhyDyn results would almost certainly result in timing the index case $t_0$ as less than 0 (too early), given 1.87 days as the top end of the tMRCA credible interval. This potentially early estimate of $t_0$ may be due to the "push-of-the-past" effect[29], which results from the assumption of deterministic dynamics in the inference process when the underlying population dynamics are stochastic (and conditioned on the persistence of a lineage). This "push-of-the-past" effect is usually reflected in an overestimate of the growth rate (or an overestimate in $R_0$) in coalescent-based inference approaches that are applied to datasets with small population sizes during their exponential growth phase[16]. Here, because $R_0$ controls not only the rate of increase in the number of infected individuals at the start of the simulated epidemic but also

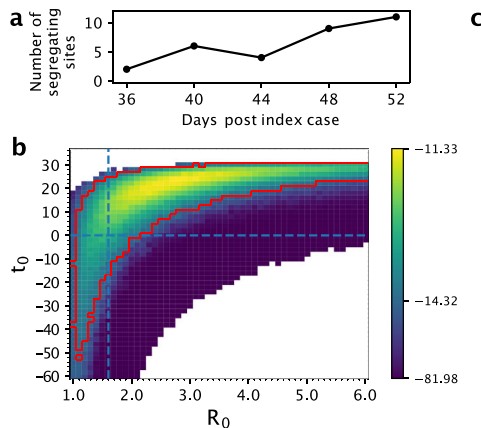

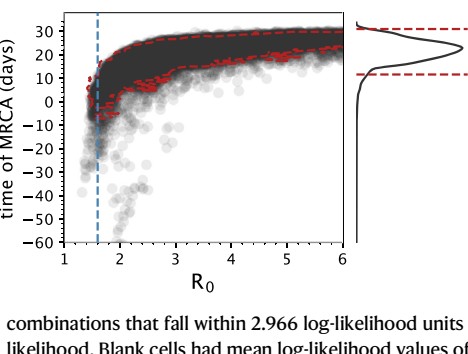

**Fig. 5 | Joint estimation of the basic reproduction number ($R_0$) and the timing of the index case ($t_0$) using early samples from the simulation, with comparison against PhyDyn. a** Simulated trajectory of the number of segregating sites using early sequences. Sequences were binned into 4-day windows, with 10 individuals sampled from each time window. **b** The log-likelihood surface based on a segregating site trajectory shown in panel (**a**). As in Fig. 3a, the log-likelihood value shown in each cell is the mean log-likelihood value calculated from 20 SMC simulations and the 95% CI boundary shown in red contains sets of parameter combinations that fall within 2.966 log-likelihood units of the maximum log-likelihood. Blank cells had mean log-likelihood values of negative infinity. (**c**) Joint density plot for $R_0$ and the time of the most recent common ancestor (tMRCA), as estimated using PhyDyn[6] on the same set of 50 sampled sequences. Dashed red line in the joint density plot shows the 95% HPD interval of the joint density. For $R_0$, only the lower bound of the 95% HPD is shown as the upper bound is above 6. In panels **a** through **c**, simulations were parameterized with a per genome, per transmission mutation rate of $\mu = 0.2$.

the time at which the simulated epidemic starts to decline, the "push-of-the-past" effect may instead be reflected in a tMRCA estimate that likely occurs too early. Because our inference approach implements stochastic population dynamics, it appropriately accounts for the push-of-the-past effect, as do phylodynamic inference approaches that incorporate stochastic population dynamics (e.g., birth-death models).

Because the impetus for developing the segregating sites inference approach was based on the extent of phylogenetic uncertainty present early on in an epidemic, we re-applied the inference approach to sequences sampled early on during the simulated epidemic, with time window bins ending on days 36, 40, 44, 48, and 52 (Fig. 5a). During each of these five-time windows, we sampled 10 sequences, resulting in a total of 50 sampled sequences. Our results on this subset of simulated data indicate that $R_0$ and $t_0$ could again be jointly estimated, although the confidence intervals for $R_0$ and $t_0$ were both considerably broader, as expected with a much shorter time series (Fig. 5b). Similarly, on this same subset of data, PhyDyn's 95% credible intervals were considerably broader (95% credible interval for $R_0 = 1.48$ to 10.80). For this particular time series, both the segregating sites approach and PhyDyn tended to overestimate the true value of $R_0 = 1.6$ (Figs. 5b, 5c). For PhyDyn, the "push-of-the-past" effect[29] may have contributed to the overestimation of $R_0$.

To determine whether there might be an upwards bias in the estimation of $R_0$ using the segregating sites approach, we simulated an additional short dataset under the same epidemiological model structure and model parameterization, with the exception of the mutation rate $\mu$, which we increased from 0.2 to 0.4. To calculate the segregating sites trajectory, we sampled from this simulation as we did for Fig. 5a–c, with 10 sequences sampled in each of the five time windows (Figure S8a). The maximum likelihood estimates of $R_0$ using our segregating sites approach did not overestimate the true $R_0$ of 1.6 in this dataset, although the time of the index case was again estimated to be slightly later than the true value of $t_0 = 0$ (Figure S8b). Compared to the results on the $\mu = 0.2$ short dataset (Fig. 5b), the 95% confidence region spanned over a similar extent of parameter space. PhyDyn also did not overestimate $R_0$ on this $\mu = 0.4$ short dataset (Figure S8c). Moreover, its 95% credible interval was considerably smaller than on the $\mu = 0.2$ short dataset. This result makes sense: at higher mutation rates, phylogenetic uncertainty is reduced and tree-based inference approaches are expected to improve. In contrast, a low-dimensional

summary statistic, such as the number of segregating sites cannot take advantage of the higher-dimensional structure present in the sequence data.

## Epidemiological inference using SARS-CoV-2 sequences from France

We applied the segregating sites inference approach to a set of SARS-CoV-2 sequences sampled from France between January 23, 2020, and March 17, 2020 (the date on which a country-wide lockdown began). We decided to apply our approach to this set of sequences for several reasons. First, many of the 479 available full-genome sequences from France over this time period appear to be genetically very similar to one another[30], indicating that one major lineage took off in France (or at least, that most sampled sequences derived from one major lineage). This lineage would be the focus of our analysis. Second, an in-depth epidemiological analysis previously inferred $R_0$ for France prior to the March 17 lockdown measures that were implemented[31]. That analysis fit a compartmental infectious disease model to epidemiological data that included case, hospitalization, and death data. Because our segregating sites inference approach can accommodate epidemiological model structures of arbitrary complexity, we could adopt the same model structure as in this previous analysis. We could also set the epidemiological parameters that were assumed fixed in this previous analysis to their same values. By controlling for model structure and the set of model parameters assumed as given, we could ask to what extent sequence data corroborate the $R_0$ estimates arrived at from detailed fits to epidemiological data.

To apply our segregating sites approach to the viral sequences from France, we first identified the subset of the 479 sequences that constituted a single, large lineage. To keep with the "tree-free" emphasis of our approach, we identified this subset of sequences ($n = 432$) without inferring a phylogeny (Methods). Using phylogenetic inference, however, we confirmed that our subset of sequences constituted a single clade, with sequences from France falling outside of this clade being excluded (Figure S9). To generate a segregating site trajectory from these sequences, we defined 4-day time windows such that the last time window ended on March 17, 2020. Figure 6a shows the number of sequences falling into each time window. Figure 6b shows the segregating site trajectory calculated from these sequences.

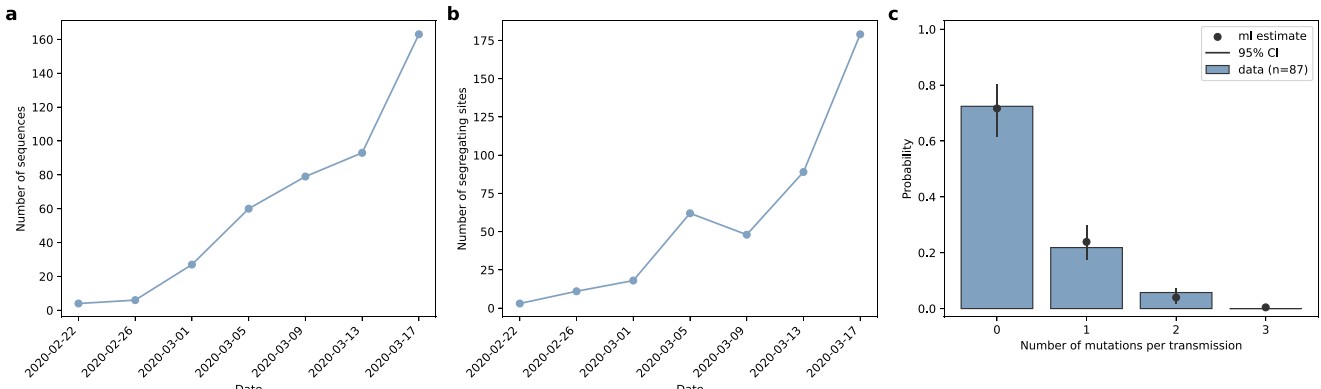

**Fig. 6 | Sequences and parameters used for epidemiological inference based on SARS-CoV-2 sequences from France. a** The number of sequences sampled over time, calculated using a 4-day time window. **b** The segregating site trajectory calculated from the binned sequences shown in panel (**a**). **c** Estimation of the per-genome, per-transmission mutation rate μ. The histogram shows the fraction of 87 analyzed transmission pairs with consensus sequences that differ from one another by the number of mutations shown on the x-axis. The mean number of mutations per transmission is μ = 0.33 (95% CI = 0.22–0.48). Black dots represent the probability of observing 0, 1, 2, and 3 mutations assuming a Poisson distribution with a mean of 0.33. Vertical black error bars span the probability of observing 0, 1, 2, and 3 mutations assuming Poisson distributions with mean values of 0.22 and 0.48.

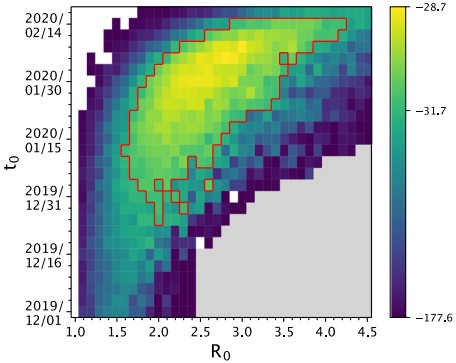

**Fig. 7 | Joint estimation of the basic reproduction number $R_0$ and the time of the index case $t_0$ for the France SARS-CoV-2 data.** The joint log-likelihood surface based on the estimated segregating site trajectory for the France data. Each cell shows the mean log-likelihood value based on 10 SMC simulations. Blank cells indicate mean log-likelihood values of negative infinity. Gray cells indicate where log-likelihood values were not evaluated. The red lines denote the set of parameter values that fall within the 95% confidence interval. A few 'islands' of parameter combinations that fall either outside or inside the 95% CI are apparent and are due to the variation in the log-likelihood values obtained from the SMC simulations.

We parameterized the model with a per genome, per transmission mutation rate $\mu$ using consensus sequence data from established SARS-CoV-2 transmission pairs that were available in the literature[32–35] (Methods). Specifically, for each of the 87 transmission pairs we had access to, we calculated the nucleotide distance between the consensus sequence of the donor sample and that of the recipient sample and fit a Poisson distribution to these data (Fig. 6c). Using this approach, we estimated a $\mu$ value of 0.33, corresponding approximately to one mutation occurring every 3 transmission events.

Similar to the approach we undertook with our simulated data, we first attempted to jointly estimate $R_0$ and the timing of the index case $t_0$ for this segregating site trajectory. We considered a broad parameter space over which to calculate log-likelihood values. Specifically, we considered $R_0$ values between 1.0 and 4.5 and $t_0$ values of between December 1st, 2019 and February 14th, 2020. We ran 10 SMC simulations and calculated the mean log-likelihood for each parameter combination (Fig. 7). We estimated $R_0$ to be 3.0 (95% confidence interval = 1.6 to 4.2), consistent with the $R_0$ estimate of 2.9 (95% confidence interval = 2.81 to 3.01) arrived at through epidemiological time

series analysis[31]. We estimated $t_0$ to be February 8th, 2020 (95% confidence interval = December 25, 2019, to February 14, 2020).

We decided to further consider an alternative model that allowed for multiple introductions of the focal lineage into France (Methods). This decision was based on evolutionary analyses that have shown that regional SARS-CoV-2 epidemics in Europe (as well as in the United States) were initiated through multiple introductions rather than only a single one[36]. Instead of attempting to jointly estimate $R_0$ and $t_0$, we attempted to jointly estimate $R_0$ and a parameter $\eta$ using the segregating site trajectory. The parameter $\eta$ quantifies the extent to which transmission between France and regions outside of France is reduced relative to transmission occurring within France. This model further required specification of the time at which the basal genotype evolved outside of France, which we refer to as $t_e$. We considered a broad parameter space over which to calculate log-likelihood values ($R_0$ values between 1.0 and 4.0 and $\eta$ values between $10^{-8}$ and $10^{-1}$) and three different $t_e$ values: December 24, 2019, January 1, 2020, and January 8, 2020 (Methods). At each of these $t_e$ values, we ran 10 SMC simulations and calculated the mean log-likelihood for each parameter combination (Fig. 8a–c). We estimated $R_0$ to be 2.6 (95% CI = 2.0 to 4.0), 2.7 (95% CI = 2.0 to 4.0), and 2.3 (95% CI = 2.1 to 4.0), respectively, under $t_e$ = December 24, 2019, January 1, 2020, and January 8, 2020. These results indicate that the inferred $R_0$ values are relatively insensitive to the assumed emergence time of the basal genotype outside of France. At later assumed values of $t_e$, our estimates for $\eta$ were higher, indicating that later emergence times were compensated for by a higher transmission rate between infected individuals outside of France and susceptible individuals within France.

We reconstructed the unobserved state variables for the multiple-introductions model using SMC simulations parameterized with $R_0$ and $\eta$ values that were sampled from the parameter spaces shown in Fig. 8, using the same approach we used for reconstructing state variables on the mock segregating sites trajectory. These reconstructed variables are shown in Fig. 9. As expected for an epidemic with an $R_0 > 1$, the total number of infected individuals increased exponentially over the time period considered (Fig. 9d–f). In Fig. 9g–i, we plot the reconstructed cumulative number of recovered individuals over time. These cumulative trajectories indicate that by mid-March 2020, approximately 0.009% to 2.044% of individuals in France had recovered from infection from this SARS-CoV-2 lineage. These cumulative predictions can be roughly compared against findings from a serological study that was conducted over this time period in France[37]. Based on a survey of 3221 individuals, this study found that 0.41% of

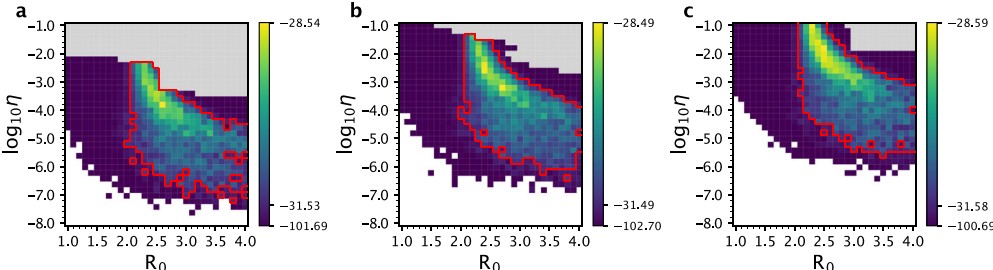

**Fig. 8 | Joint estimation of the basic reproduction number $R_0$ and the transmission-reduction parameter $\eta$ for the multiple-introductions model using the France data.** The joint log-likelihood surface based on the estimated segregating site trajectory for the France data is shown under three different basal genotype emergence times: $t_e$ = December 24, 2019 (**a**), January 1, 2020 (**b**), and January 8, 2020 (**c**). Each cell shows the mean log-likelihood value based on 10 SMC simulations. Blank cells indicate mean log-likelihood values of negative infinity. Gray cells indicate where log-likelihood values were not evaluated due to extended simulation time. The red lines in each panel denote the set of parameter combinations that fall within the 95% confidence interval. As in Fig. 7, a few 'islands' of parameter combinations are apparent due to the variation in the log-likelihood values obtained from the SMC simulations.

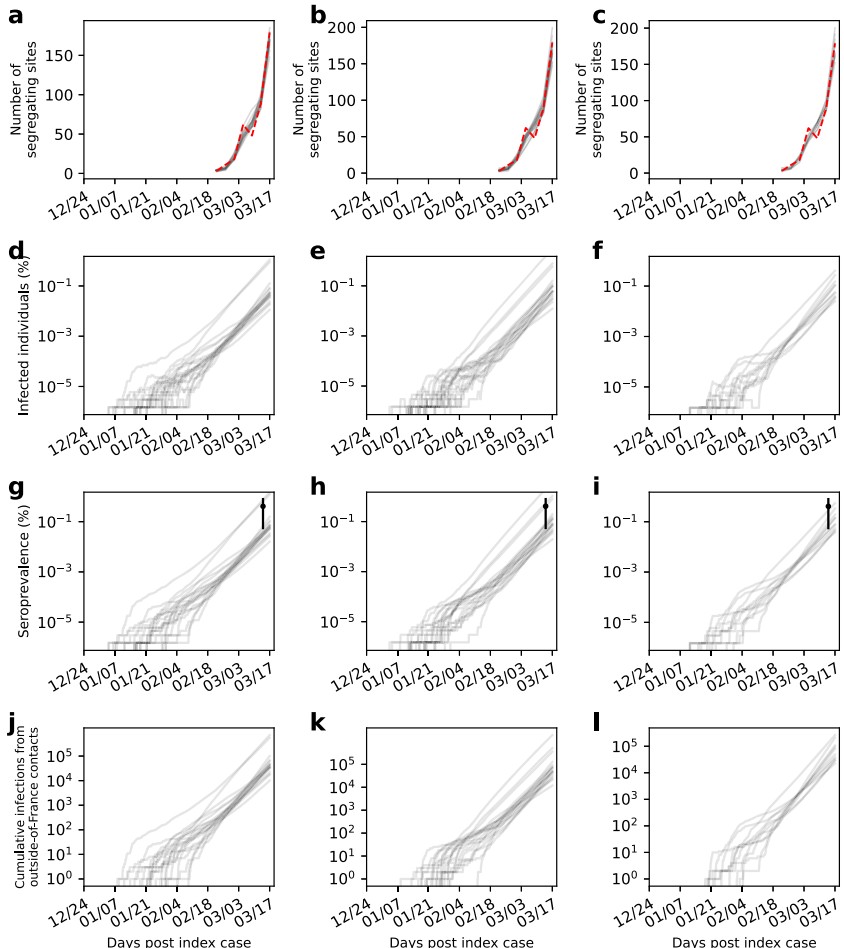

**Fig. 9 | Trajectories of reconstructed state variables for the France data under the multiple-introductions model.** State variables are reconstructed for the multiple-introductions model with three different values assumed for the emergence time of the basal genotype: $t_e$ = December 24, 2019 (first column), January 1, 2020 (second column), and January 8, 2020 (third column). **a–c** Segregating site trajectory for the France SARS-CoV-2 data (red), alongside reconstructed segregating site trajectories (gray). **d–f** Reconstructed dynamics of the number of infected individuals ($E_1 + E_2 + I$) over time, shown in percent of France's population. **g–i** Reconstructed dynamics of the cumulative number of recovered individuals over time, shown in percent of France's population. Independent estimates of the fraction of the population that has been infected with SARS-CoV-2 by mid-March are shown in black. Estimates are from a serological study conducted during the time window March 9-15, 2020[37]. **j–l** Reconstructed dynamics of the cumulative number of infections in France that resulted from contact with infected individuals outside of France. Reconstructed state variables shown in panels (**a–l**) were obtained by running the particle filter using $R_0$ and $t_0$ parameter values randomly sampled from within the 95% CI region, with a further condition that the log-likelihood from the run exceeded the 95% CI region log-likelihood cutoff shown in Fig. 8a–c, respectively.

individuals (95% confidence interval = 0.05% to 0.88%) had gotten infected with SARS-CoV-2 by March 9 to 15, 2020 (Fig. 9g–i). Our estimates fall in line with these independent estimates. Of note, our estimates should fall on the low side of these independent estimates because other, smaller clades were also circulating in France during the time period studied and infections with viruses from these other clades would also contribute to seropositivity levels. We also emphasize that this is necessarily a rough comparison because seroconversion does not occur exactly at the point of recovery. It can occur over a broader range of times, ranging from prior to recovery to many days following symptom onset[38]. Finally, in Fig. 9j–l, we plotted the reconstructed cumulative number of infections that resulted directly from contact with individuals outside of France. By the first sampled time window (ending on February 22, 2022), our SMC results indicate that there were very likely repeated introductions of this lineage into France, with the majority of sampled particles pointing towards hundreds of introductions of this lineage into France by this time point.

## Discussion

Here, we developed a statistical inference approach to estimate epidemiological parameters from virus sequence data. Our inference approach is a "tree-free" approach in that it does not rely on the reconstruction of viral phylogenies to estimate model parameters. One benefit of using such an approach for parameter estimation of emerging viral pathogens is that, early on in an epidemic, phylogenetic uncertainty present in time-resolved viral phylogenies is significant, and tree-based phylodynamic inference approaches would need to integrate over this uncertainty. This is oftentimes computationally intensive, especially when many sequences have been sampled. The computational complexity of our "tree-free" approach, in contrast, does not scale with the number of sampled sequences. Instead, the runtime required for parameter inference depends on the number of genotypes that evolve over the course of the model simulations. This number in turn is affected by the proposed basic reproduction number, the proposed time of the index case in the single introduction model, and the magnitude of the per genome, per transmission mutation rate μ. A second benefit to our tree-free approach is that it can estimate the time of the index case (in a single-introduction scenario), whereas tree-based inference methods estimate the time of the most recent common ancestor. This is a benefit when the question of interest focuses on when a viral lineage emerges and starts to spread. Instead of viral phylogenies being the data that statistically interface with the epidemiological models, our approach uses a population genetic summary statistic of the sequence data, namely the number of segregating sites present in time-binned sets of viral sequences. Our inference approach benefits from being plug-and-play in that it can easily accommodate different epidemiological model structures.

Based on fits to a simulated data set, we have shown that segregating site trajectories can be used to estimate the basic reproduction number $R_0$ and the timing of the index case $t_0$ in cases where a single introduction can be assumed. We further fit a multiple-introductions epidemiological model to a segregating site trajectory that was calculated from SARS-CoV-2 sequence data from France, estimating a basic reproduction number $R_0$ of approximately 2.3-2.7. These results are consistent with previous estimates from an epidemiological analysis and consistent with a serological study conducted in mid-March 2020.

Our inference approach relies on several assumptions that are shared by existing phylodynamic inference methods. Most notably, it relies on an assumption that all mutations are phenotypically neutral. However, a recent analysis of SARS-CoV-2 sequences has shown evidence for purifying selection, even early on during the pandemic[39]. Indeed, within the set of SARS-CoV-2 sequences from France, we observe 170 nonsynonymous mutations and 138 synonymous mutations (a ratio of 1.23:1). Given the number of nonsynonymous sites

($n = 68,540$) and the number of synonymous sites ($n = 19,255$) in the SARS-CoV-2 genome, we would expect, under neutrality, a ratio of 3.56:1. This underrepresentation of nonsynonymous genetic variation points towards purifying selection in our analyzed dataset. A more recent analysis also raises the possibility of adaptive evolution occurring during early 2020[40]. Incorporating non-neutral genetic variation into inference approaches such as ours and existing phylodynamic ones is complicated, although some statistical approaches have started to tackle this goal[9]. In the context of our segregating sites inference approach, directly incorporating non-neutral evolution will increase model complexity considerably, and assumptions would need to be made about the distribution of mutational fitness effects. Rather than incorporating non-neutral evolution within our approach, we can for now consider how the occurrence of non-neutral evolution would impact our parameter estimates. With purifying selection at play, we would expect to see less genetic variation than in its absence. As such, the number of segregating sites in any time window would be lower than it would be under neutrality. Our inference approach, assuming neutrality, would therefore bias $R_0$ estimates to be low and, in single-introduction models, the timing of the index case $t_0$ to be late. In multi-introduction models, our estimate of $\eta$ would be biased high.

Our approach also assumes infinite sites and the absence of homoplasies. While these assumptions are limiting over longer periods of sequence evolution, our approach is intended to be used for emerging viral pathogens, sampled over shorter periods of time, when levels of genetic diversity are still low. As such, these assumptions will likely not be violated in cases where this approach will come in useful. We would also like to note that the infinite sites assumption could in principle be relaxed, but this would make the simulations in the inference approach substantially more costly. Furthermore, as time goes on, not only do chances of repeated mutations at sites increase, but genetic diversity increases. As such, phylogenetic uncertainty will decrease, such that existing tree-based phylodynamic inference approaches will become increasingly informative and segregating site trajectories less informative.

While our inference approach does adopt assumptions of phenotypic neutrality and infinite sites, it does not assume a constant sampling rate or a specific sampling process throughout the time period over which sequences are collected. As we have shown in Fig. 1b, sampling effort does impact the segregating sites trajectory: the greater the sampling effort, the larger the number of segregating sites. For our inference approach to perform effectively, sampling effort therefore needs to be matched between the simulations and the empirical data. This matching of sampling effort is implemented in the particle filter. However, the number of samples sequenced per time window is not particularly informative of model parameters (except in the case of extremely high sampling effort when certain low $R_0$ model parameterizations cannot appropriately evaluate the expected number of segregating sites in a time window because the number of sampled sequences exceed the number of simulated recoveries). The reason why the number of samples is not particularly informative of model parameters is because, under our approach, sampling of individuals does not impact the underlying epidemiological dynamics: individuals are sampled upon recovery, once they are no longer infectious. That the number of observed samples is not highly informative of model parameters we see as a benefit of our approach because sampling effort and testing rates can change dramatically over the course of an emerging pandemic or over the early period of an emerging viral lineage as surveillance efforts ramp up. In contrast, sampling times of sequences have been shown to be highly informative of model parameters in the case of birth-death models, with sampling process misspecification resulting in the possibility of arriving at biased parameter estimates[41].

While the number of sampled sequences is largely uninformative of model parameters, our approach does have to make an assumption

of when individuals are sampled. In our simulated dataset and in our application to SARS-CoV-2, we assumed that individuals were sampled as they recovered. This sampling scheme decision was based on our understanding that the time of symptom onset often follows peak viral load for many emerging viral pathogens[42] and an assumption that most testing early on in a pandemic involves individuals who develop symptoms. It is important to note that if the assumed sampling scheme is mismatched with the empirical sampling scheme, parameter estimates may be biased. For example, if individuals were instead sampled as they transitioned from the exposed class to the infectious class, rather than upon recovery, and we assumed in our model that individuals were sampled upon recovery, then our $R_0$ estimates would be biased high.

Finally, we would like to note that setting the per genome, per transmission mutation rate to a constant value does not correspond to an assumption of a constant molecular clock. A constant molecular clock requires that the number of substitutions per unit time remains the same. Our assumption is that the mean number of nucleotide changes that occur during a transmission event between a donor and a recipient (at the consensus level) stays constant over time. This would almost certainly be the case unless the fidelity of the viral polymerase was evolving over the period considered. Changes in the substitution rate could come about if the generation interval between transmission events changes due, for example, to the implementation of non-pharmaceutical interventions or increased symptom awareness. A shortening of the generation interval (defined as the time between infection and onward transmission) would increase the number of transmission events that occur per unit time and thereby result in an increase in the substitution rate. In contrast, a lengthening of the generation interval would result in fewer transmission events occurring per unit time, thereby decreasing the population-level substitution rate. Changes in the generation interval can emerge from an underlying epidemiological model, such that our assumption of a constant per genome, per transmission event mutation rate does not preclude or conflict with the observation of changes in the substitution rate over time.

The analysis we presented here focuses on statistical inference using sequence data alone. In recent years, there has also been a growing interest in combining multiple data sources – for example, sequence data and epidemiological data or serological data - to more effectively estimate model parameters. The few existing studies that have incorporated additional data while performing phylodynamic inference have shown the value in pursuing this goal[7,43,44]. As a next step, we aim to extend the segregating sites approach developed here to incorporate epidemiological data and/or serological data more explicitly. Straightforward extension is possible due to the state-space model structure that is at the core of the particle filtering routine we use.

Our analysis focused on phylodynamic inference based on sequence data belonging to a single viral lineage, with either a single index case or multiple introductions from an outside reservoir. Our approach, however, can be expanded in a straightforward manner to multiple viral lineages. This is especially useful in cases like SARS-CoV-2, where many regions have witnessed the introduction of multiple clades[10,45]. In this case, a single segregating sites trajectory could be calculated for each clade, such that multiple segregating site trajectories could be simultaneously fit to under specified constraints such as the basic reproduction number being the same across all clades. Different clades could also be allowed to differ in their reproductive numbers, such that questions relating to the selective advantage of some clades over others could be addressed. As such, this inference method, designed for emerging pathogens with low levels of genetic diversity, may continue to be useful for endemic pathogens to address questions related to the emergence of new viral lineages.

## Methods

### Brief overview of inference approach

Mutations occur during viral replication within infected individuals and these have the potential to be transmitted. During the epidemiological spread of an emerging virus or viral lineage, the virus population (distributed across infected individuals) thus accrues mutations and diversifies genetically. This joint process of viral spread and evolution can be simulated forward in time using compartmental models, with patterns of epidemiological spread leaving signatures in the evolutionary trajectory of the virus population. Parameters of these compartmental models that govern patterns of epidemiological spread can thus in principle be estimated using viral sequence data. Here, similar in spirit to existing inference approaches based on summary statistics[46–50], we develop a statistical inference approach that fits compartmental epidemiological models to time series of a low-dimensional summary statistic calculated from sequence data. Specifically, we use trajectories of the number of segregating sites from samples of the viral population taken over time for statistical inference. Because we propose the use of our method early on in an epidemic (or during the early expansion of a viral lineage), we focus primarily on estimating the basic reproduction number $R_0$ using this inference approach.

### Epidemiological model simulations and calculation of segregating site trajectories

To simulate mock data of segregating site trajectories, we specify a compartmental epidemiological model and simulate the model under demographic stochasticity using Gillespie's τ-leap algorithm. Here, we use a susceptible-exposed-infected-recovered (SEIR) model whose stochastic dynamics are governed by the following equations:

$$S_{t+\Delta t} = S_t - n_{S \to E} \tag{1}$$

$$E_{t+\Delta t} = E_t + n_{S \to E} - n_{E \to I} \tag{2}$$

$$I_{t+\Delta t} = I_t + n_{E \to I} - n_{I \to R} \tag{3}$$

$$R_{t+\Delta t} = R_t + n_{I \to R} \tag{4}$$

where:

$$n_{S \to E} \sim Pois\left(\beta \frac{S_t}{N} I_t \Delta t\right) \tag{5}$$

$$n_{E \to I} \sim Pois(\gamma_E E_t \Delta t) \tag{6}$$

$$n_{I \to R} \sim Pois(\gamma_I I_t \Delta t) \tag{7}$$

where $\beta$ is the transmission rate, $N$ is the host population size, $\gamma_E$ is the rate of transitioning from the exposed to the infected class, $\gamma_I$ is the rate of recovering from infection, and $\Delta t$ is the τ-leap time step used. $R_0$ is given by $\beta / \gamma_I$. The epidemiological dynamics of this model can be simulated from the above equations alone. Additional complexity is needed to incorporate virus evolution throughout the course of the simulation. To incorporate virus evolution, we partition exposed individuals and infected individuals into genotype classes, with genotype 0 being the reference genotype present at the start of the simulation. Mutations to the virus occur at the time of transmission, with the number of mutations that occur in a single transmission event given by a Poisson random variable with mean $\mu$, the per-genome, per-transmission event mutation rate. We assume infinite sites such that new mutations necessarily result in new genotypes. New mutations

and new genotypes are both assigned integer indices in order of their appearance. When new mutations are generated at a transmission event, the new genotype harbors the same mutation(s) as its parent genotype plus any new mutations. We use a sparse matrix approach to store genotypes and their associated mutations to save on memory.

There are three types of events that occur in the SEIR model simulations: transitions from exposed to infected; transitions from infected to recovered; and transmission. To simulate transitions from exposed to infected, during a time step $\Delta t$, $n_{E \to I}$ individuals are drawn at random from the set of individuals who are currently reside in the exposed class. These individuals will transition to the infected class during this time step, while retaining their current genotype statuses. To simulate transitions from infected to recovered, during a time step $\Delta t$, $n_{I \to R}$ individuals are drawn at random from the set of individuals who are currently residing in the infected class. These individuals will transition to the recovered class during this time step. To simulate transmission, during a time step $\Delta t$, we add $n_{S \to E}$ new individuals to the set of exposed individuals. For each newly exposed individual, we randomly choose (with replacement) a currently infected individual as its 'parent'. If no mutations occur during transmission, then this newly exposed individual enters the same genotype class as its parent. If one or more mutations occur during transmission, this newly exposed individual enters a new genotype class, and the sparse matrix is extended to document the new genotype and its associated mutations (given as integers, without a bitstring or explicit genome structure).

We start the simulation with one infected individual carrying a viral genotype that we consider as the reference genotype (genotype 0). To calculate a time series of segregating sites, we define a time window length T ($T > \Delta t$) of a certain number of days and partition the simulation time course into discrete, non-overlapping time windows. During simulation, we keep track of the individuals that recover (transition from $I$ to $R$) within a time window. For each time window $i$, we then sample $n_i$ of these individuals at random, where $n_i$ is the number of sequences sampled in a given time window based on the sampling scheme chosen. Because we have the genotypes of the sampled individuals from the sparse matrix, we can calculate the number of segregating sites $s_i$ in any time window $i$. Since $s_i$ is the number of polymorphic sites across the sampled individuals in time window $i$, it is simply calculated from the set of mutations harbored by the sequences of the sampled individuals. While in our simulations, we sample individuals as they recover, alternative sampling schemes can instead be assumed. For example, individuals could be sampled as they transition from the exposed to the infected class, or while they are in the infected class. We chose to sample upon recovery based on symptom development (and thereby testing) often occurring following peak viral load.

### Implementation of the transmission heterogeneity model
We implement transmission heterogeneity in the epidemiological model by splitting the infected classes into a high-transmission and a low-transmission class, as has been done elsewhere[6,10]. For an SEIR model, the model extended to incorporate transmission heterogeneity becomes:

$$S_{t+\Delta t} = S_t - n_{S \to E} \tag{8}$$

$$E_{t+\Delta t} = E_t + n_{S \to E} - n_{E \to I_h} - n_{E \to I_l} \tag{9}$$

$$I_{h,t+\Delta t} = I_{h,t} + n_{E \to I_h} - n_{I_h \to R} \tag{10}$$

$$I_{l,t+\Delta t} = I_{l,t} + n_{E \to I_l} - n_{I_l \to R} \tag{11}$$

$$R_{t+\Delta t} = R_t + n_{I_h \to R} + n_{I_l \to R} \tag{12}$$

where:

$$n_{S \to E} \sim Pois\left(\beta_h \frac{S_t}{N} I_{h,t} \Delta t\right) + Pois\left(\beta_l \frac{S_t}{P} I_{l,t} \Delta t\right) \tag{13}$$

$$n_{E \to I} \sim Pois(\gamma_E E_t \Delta t) \tag{14}$$

$$n_{E \to I_h} \sim Bin(n_{E \to I}, p_H) \tag{15}$$

$$n_{E \to I_l} = n_{E \to I} - n_{E \to I_h} \tag{16}$$

$$n_{I_h \to R} \sim Pois(\gamma_I I_{h,t} \Delta t) \tag{17}$$

$$n_{I_l \to R} \sim Pois(\gamma_I I_{l,t} \Delta t) \tag{18}$$

The parameter $p_H$ quantifies the proportion of exposed individuals who transition to the high-transmission $I_h$ class. Parameters $\beta_h$ and $\beta_l$ quantify the transmission rates of the infectious classes that have high and low transmissibility, respectively. We set the values of $\beta_h$ and $\beta_l$ based on a given parameterization of overall $R_0$ and the parameter $p_H$. To do this, we first define, as in previous work[6,10], the relative transmissibility of infected individuals in the $I_h$ and $I_l$ classes as $c = \frac{\beta_h}{\beta_l}$. We further define a parameter $P$ as the fraction of secondary infections that result from a fraction $p_H$ of the most transmissible infected individuals. Based on given values of $p_H$ and $P$, we set $c$, as in previous work[10], to $\frac{[\frac{1-p_H}{p_H}]}{[\frac{1}{P}-1]}$. With $c$ defined in this way, $p_H$ can be interpreted as the proportion of most infectious individuals that result in $P$ of secondary infections. We set $P$ to 0.80, to make $p_H$ easily interpretable relative to the "20/80" rule in disease ecology[22]. Recognizing that $R_0 = \frac{p_H \beta_h + (1-p_H)\beta_l}{\gamma_I}$ in this model, we can then solve for $\beta_l$: $\frac{R_0 \gamma_I}{p_H c + (1-p_H)}$, and set $\beta_h = c\beta_l$. Note that the interpretation of $p_H$ in the context of the disease ecology rule is an approximation since this calculation does not take into consideration variation in individual $R_0$ that results from differences in the duration of infection or variation in individual $R_0$ that results from differences in the number of secondary infections that are due to stochastic effects.

### Epidemiological inference using time series of segregating sites
Our inference approach relies on particle filtering, also known as Sequential Monte Carlo (SMC), to estimate model parameters and reconstruct unobserved (latent) state variables. Particle filtering calculates the likelihood of a parameterized model (more precisely, the probability of observing the time-series data marginalized over the unobserved state variables) by recurrently updating a set of particles (Figure S10). In our case, each of these particles holds a state-space model, which includes a process model component that simulates underlying epidemiological and evolutionary dynamics and an observation model that relates these latent state variables to the observed segregating sites data (Figure S11). The process model includes the unobserved epidemiological variables (e.g., $S$, $E$, $I$, and $R$) and the evolutionary components of the model (viral genotypes and mutations). From one observed segregating sites data point to the next one, the model is simulated using Gillespie's $\tau$-leap algorithm, as described in the section above.

At the end of each time window, when the simulation reaches the next observed segregating sites data point, the observation model is used to calculate the probability of observing the observed data point given the underlying process model. This probability is calculated as follows. We calculate the expected number of segregating sites from

the model simulation by performing $k$ 'grabs' of sampled individuals, with each grab consisting of the following steps:

- pick (without replacement) $n_i$ individuals from the set of individuals who recovered during time window $i$, where $n_i$ is the number of samples present in the empirical dataset in window $i$. This step mimics the process of sample collection at the same effort as in the observed data. We control for sampling effort because the extent of sampling impacts the number of segregating sites.
- calculate the simulated number of segregating sites $s_i^{sim}$, based on the genotypes of the sampled $n_i$ individuals.

Between grabs, the replacement of previously sampled individuals occurs. We then calculate the mean number of segregating sites for window $i$ by taking the average of all $k$ $s_i^{sim}$ values. Finally, we calculate the probability of observing $s_i$ segregating sites in window $i$, given the model-simulated mean number of segregating sites, using a Poisson probability mass function parameterized with the mean $s_i^{sim}$ value and evaluated at $s_i$. As a special case in the calculation of this probability, if the number of individuals who recovered during a given time window $i$ is less than the number that needs to be sampled ($n_i$), then the particle's probability of observing the number of segregating sites $s_i$ is set to 0. The calculated probabilities serve as the weights for the particles.

Particle weights obtained at the end of each window are used 1) to resample particles for the next time window according to their assigned weights and 2) to calculate the likelihood of a parameterized model. In the particle filtering algorithm, the likelihood is obtained by averaging particle weights within each window and then multiplying these average particle weights across all time windows with observations. For time windows without observations ($n_i = 0$), particle weights are assigned a value of 0 if the virus has died out stochastically and 1 if the virus continues to persist in the population. These weights are used for resampling, but do not contribute to the calculation of the likelihood. We adopt this approach to filter out particles during early time windows that have undergone stochastic extinction.

Latent state variables are reconstructed by randomly sampling a particle at the end of an SMC simulation and plotting the values of its simulated latent state variables over time. All of our SMC simulations were performed with 200 particles and $k = 50$ grabs. Note that the complexity of this inference method is largely independent of the number of input sequences. This stands in contrast to phylodynamic inference approaches that frequently down-sample sequences to reduce runtime.

### Converting simulated sequences into nucleotide sequences for the performance comparison against PhyDyn

Simulated sparse matrices were converted to nucleotide alignments by first generating a reference sequence with the same length as the maximum number of mutations in the sparse matrix and choosing an A, C, G, or T nucleotide at each site with equal probability. A mutated sequence was generated for each genotype represented in the sparse matrix by replacing the reference allele at that position with another nucleotide chosen with equal probability. The final FASTA alignment was generated by identifying the simulated sequence associated with each sampled individual. Generation of the simulated FASTA file was done using Python v3.9.4 with Numpy v1.19.4.

The simulated FASTA alignment was used to generate a BEAST2 XML file from a template XML which was generated in part using Beauti v2.6.6. This template used a JC69 nucleotide substitution model with no invariant sites. We assumed an uncorrelated log-normally distributed relaxed clock with a uniform [0.0, 1E-2] prior on the mean and a uniform [0.0, 2.0] prior on the standard deviation.

A single-deme structured coalescent prior as defined by the following equations was implemented using PhyDyn v1.3.8:

$$\frac{dE}{dt} = \frac{\beta IS}{N} - \gamma_E E \tag{19}$$

$$\frac{dI}{dt} = \gamma_E E - \gamma_I I \tag{20}$$

$$\frac{dR}{dt} = \gamma_I I \tag{21}$$

where $\beta = R_0 \gamma_I$. A population size of $10^5$ with a single initially infected individual was used. We assume infected individuals remain exposed for an average of 2 days ($1/\gamma_E$) and infectious ($1/\gamma_I$) for an average of 3 days. $R_0$ was estimated using a uniform [1.0, 10.0] prior. All sampled sequences were assigned to the infected ("I") class.

Sampled parameters and trees were logged every 1000 states and all MCMC chains were run for at least 209 M (Fig. 3b), 64 million (Fig. 5c), 150 million (Figure S8c) iterations. The first 10% of MCMC chains were discarded as burn-in and the ESS values of all parameters were >200, as identified by Tracer v1.7.1 (10.1093/sysbio/syy032).

### Epidemiological model structure and parameterization used in the SARS-CoV-2 analysis

The process model we use in our application to SARS-CoV-2 sequence data from France is based on a previously published epidemiological model[31]. We base our process model on this published model to allow for a direct comparison of inferred $R_0$ values between our sequence-based analysis and their analysis that focuses on SARS-CoV-2 spread in France over a similar time period. Their analysis was based on fitting an epidemiological model to a combination of case, hospitalization, and death data. Their model structure, once implemented using Gillespie's $\tau$-leap algorithm, is given by:

$$S_{t+\Delta t} = S_t - n_{S \to E1} \tag{22}$$

$$E_{1,t+\Delta t} = E_{1,t} + n_{S \to E1} - n_{E1 \to E2} \tag{23}$$

$$E_{2,t+\Delta t} = E_{2,t} + n_{E1 \to E2} - n_{E2 \to I} \tag{24}$$

$$I_{t+\Delta t} = I_t + n_{E2 \to I} - n_{I \to R} \tag{25}$$

$$R_{t+\Delta t} = R_t + n_{I \to R} \tag{26}$$

where:

$$n_{S \to E1} \sim Pois\left(\beta \frac{S_t}{N} I_t \Delta t\right) + Pois\left(\beta \frac{S_t}{N} E_{2,t} \Delta t\right) \tag{27}$$

$$n_{E1 \to E2} \sim Pois(\gamma_{E1} E_{1,t} \Delta t) \tag{28}$$

$$n_{E2 \to I} \sim Pois(\gamma_{E2} E_{2,t} \Delta t) \tag{29}$$

$$n_{I \to R} \sim Pois(\gamma_I I_t \Delta t) \tag{30}$$

The parameters are the transmission rate $\beta$, the rate of transitioning from the $E_1$ class to the $E_2$ class $\gamma_{E1}$, the rate of transitioning from the $E_2$ class to the $I$ class $\gamma_{E2}$, and the rate of transition from the $I$ class to the $R$ class $\gamma_I$. The average duration of time spent in the $E_1$ class given by $1/\gamma_{E1} = 4$ days, the average duration of time spent in the $E_2$ class given by $1/\gamma_{E2} = 1$ day, and the average duration of time spent in

the infected class given by $1/\gamma_I = 3$ days. Their model assumes that the transmission efficiency $\beta$ of exposed class 2 ($E_2$) and that of the infected class $I$ are the same; their model considers $E_2$ and $I$ as distinct classes to interface with the case data, where symptoms are assumed to not appear before an individual has transitioned to class $I$. We maintain the model structure with $E_1$, $E_2$, and $I$ rather than reducing it to a model structure with just a single $E$ and a single $I$ class to keep the same overall distribution of infection times as in their model.

Because SARS-CoV-2 dynamics are characterized by substantial levels of transmission heterogeneity[10,23,51] and we have shown in Fig. 1 that transmission heterogeneity impacts segregating site trajectories, we expanded the compartmental epidemiological model for SARS-CoV-2 described above to include transmission heterogeneity in a manner similar to the one we used in Fig. 1. Based specifically on the analysis by Paireau and colleagues[52], we set $p_H$ to 0.10, such that 10% of infections are responsible for 80% of secondary infections. Analogous to the approach we undertook for the simulated data, we jointly estimated $R_0$ and $t_0$ using the segregating site trajectory shown in Fig. 6b.

Based on phylogenetic analyses that have indicated that early introductions of SARS-CoV-2 into focal regions likely resulted from multiple introductions rather than a single one, we considered a modified version of the epidemiological model that would allow for multiple introductions. The modification relied on the incorporation of infections within France that resulted from direct contact with infected individuals outside of France, termed the viral "reservoir". Similar to the approach adopted by some existing phylodynamic analyses[12], the viral population dynamics in this reservoir are simplified to exponential growth. This infected population from outside of France acts as another source of infection for susceptible individuals within France, allowing for multiple introductions of SARS-CoV-2 into France.

As in the focal region, new genotypes are expected to emerge in the outside reservoir. As we assume an infinite sites model, the genotypes that emerge in the outside reservoir and in the focal region will not overlap except in the basal genotype that is first introduced to the focal region. For this reason, and because the basal genotype is expected to be considerably more common than any of the viral genotypes that stem from it, we consider only the repeated introduction of the basal genotype into France. Starting at the time of emergence of the basal genotype in the outside reservoir ($t_e$), we let the number of individuals infected with this basal genotype $Y_t$ grow exponentially:

$$Y_t = e^{r(t - t_e)} \qquad (31)$$

where $r$ is the intrinsic growth rate of the basal genotype. Based on empirical estimates[53,54], we set the intrinsic growth rate to 0.22 day$^{-1}$. To set $t_e$, we first identified the genotype sampled in France that is genetically closest to the reference strain Wuhan/Hu-1 (MN908947.3). This basal genotype differs from Wuhan/Hu-1 by 4 nucleotides: C241T, C3037T, C14408T, and A23403G. Using GISAID data, we then identified sequences with collection locations outside of France that carried all four of these mutations that define the basal genotype. The earliest of these sequences including the four basal genotype-defining mutations was collected on January 25, 2020, in Australia, suggesting that the basal genotype had been circulating prior to January 25, 2020. Considering the potential delay between emergence and the time of first detection, we considered three distinct $t_e$ values: December 24th, 2019, January 1st, 2020, and January 8th, 2020.

Individuals infected in this outside reservoir can transmit their infection to susceptible individuals within France. The rate at which they transmit the infection is reduced relative to the rate at which infected individuals within France transmit the infection to susceptible individuals within France. We let the factor by which transmission is reduced be given by the factor $\eta$. During a $\tau$-leap timestep, the number of individuals within France who become infected from contact with an infected individual outside of France is therefore given by:

$$n^O_{S \to E1} \sim Pois\left(\beta\eta \frac{S_t}{N} Y_t \Delta t\right) \qquad (32)$$

As we are considering only the transmission of the basal genotype from infected individuals in the outside reservoir to susceptible individuals in France, all of these newly infected individuals will carry the basal genotype unless mutation occurs during the transmission process. Our simplifying assumption that only the basal genotype can be introduced into France from the outside reservoir ignores the possibility that genotypes that are derived from the basal genotype enter France from the outside reservoir. Strictly speaking, we think this assumption is unlikely to be met. However, at very early time points in France's epidemic, most of the genotypes outside of France should still be the basal genotype, and only at later time points should the frequencies of derived genotypes increase outside of France. Introduction of these derived genotypes at these later time points could result in the establishment of viral sublineages in France. However, because autochthonous infections would be high at this point, these viral sublineages would very likely go unsampled. As such, we do not think that our assumption of only the basal genotype being introduced into France would have a dramatic effect on our results. We can consider, however, the effects that violation of this assumption would have on our parameter estimates: if derived genotypes were introduced into France and sampled (or their descendants sampled), then the number of segregating sites that would have evolved within France would be lower than we are currently taking it to be. As such, our current estimate of $R_0$ would be biased high.

### Estimation of the per genome, per transmission event mutation rate

We set the per-genome, per-transmission mutation rate parameter $\mu$ to 0.33. This is based on the fit of a Poisson distribution to the number of de novo substitutions that were observed in 87 transmission pairs of SARS-CoV-2 from four studies[32–35]. Accession numbers for 78/87 of these transmission pairs are available in Table S1. Accession numbers for the remaining pairs were provided by the corresponding authors of the relevant publication[34]. Sequence data were aligned to Wuhan/Hu-1 (MN908947.3)[55] using MAFFT v.7.464[56]. Insertions relative to Wuhan/Hu-1 were removed and the first 55 and last 100 nucleotides of the genome were masked. De novo substitutions for each pair were identified in Python v.3.9.4 (http://www.python.org) using NumPy v.1.19.4[57]. Ambiguous nucleotides were permissively included in the identification of de novo substitutions (e.g., an $R$ nucleotide was assumed to match both an $A$ and a $G$). The mean number of substitutions between transmission pairs is the maximum likelihood estimate for the rate parameter of the Poisson distribution. The 95% confidence interval was calculated using the exact method using SciPy v.1.5.4[58].

The value for $\mu = 0.33$ is consistent with population-level substitution rate estimates for SARS-CoV-2, which range from 7.9 ×10$^{-4}$ to 1.1 ×10$^{-3}$ substitutions per site per year[28,59]. With a genome length of SARS-CoV-2 of approximately 30,000 nucleotides and a generation interval of approximately 4.5 days[60], these population-level substitution rates would correspond to per genome, per transmission mutation rates of between 0.29 and 0.41, respectively.

### Estimation of segregating site trajectories for the France data

We downloaded all complete and high-coverage SARS-CoV-2 sequences with complete sampling dates sampled through March 17th, 2020 (https://doi.org/10.55876/gis8.230123mt) in France and uploaded through April 29th, 2021 from GISAID[61]. Sequences were aligned to Wuhan/Hu-1 using MAFFT v.7.464. Insertions relative to Wuhan/Hu-1 were removed. Any sequences with fewer than 28000 $A$, $C$, $T$, or $G$

characters were removed. Following this filtering protocol, our dataset included 479 sequences. We masked the first 55 and last 100 nucleotides in the genome as well as positions marked as "highly homoplasic" in early SARS-CoV-2 sequencing data (https://github.com/W-L/ProblematicSites_SARS-CoV2/blob/master/archived_vcf/problematic_sites_sarsCov2.2020-05-27.vcf). Pairwise SNP distances were calculated in a manner that accounted for IUPAC ambiguous nucleotides in Python using NumPy. To subset these data to a single clade circulating within France, we identified the connected components of this pairwise distance matrix with a cutoff of 1 SNP in Python using SciPy and identified the shared SNPs relative to Wuhan/Hu-1 between all sequences in each connected component. The largest connected component contained 308 sequences which shared the substitutions C241T, C3037T, C14408T, and A23403G. Our final dataset included these 308 as well as 124 sequences from connected components that shared these four substitutions relative to Wuhan/Hu-1. We included connected components in which all sequences had an *N* at any of the four clade-defining sites of the largest connected component. Two sequences were excluded as they differed from all other sequences in the dataset by > 7 SNPs. This dataset includes 112 of the 186 sequences analyzed in Danesh et al.[11]. Sequences were binned into four-day windows, aligned such that the last window ended on the latest sampling date. The number of segregating sites in each window was calculated in Python using NumPy. Ambiguous nucleotides were permissively considered in the calculation of segregating sites, e.g., an *N* nucleotide was assumed to match all four nucleotides, whereas an *R* nucleotide was assumed to match only *A* and *G* nucleotides. This matching assumption results in a lower bound estimate for the number of segregating sites in any time window. If we instead count an *N* nucleotide at a site as a mutation, the number of segregating sites in each time window is much larger (Figure S12a). However, it is unlikely that an *N* nucleotide indicates a mutation; it is much more likely that an *N* indicates an inability to call a nucleotide based on low read depth or poor quality scores at a *site*. If we count *N* nucleotides as matching observed variation but count other ambiguous nucleotides (e.g., *R*) as mutations, the segregating site trajectory is barely affected (Figure S12b). This is because there are very few non-*N* ambiguous nucleotides in the dataset. As such, our parameter estimates on the France dataset are unlikely to be impacted by our assumption of ambiguous nucleotides matching observed genetic variation at their respective sites.

**Phylogenetic analysis of SARS-CoV-2 sequences from France**
To confirm that the subset of sequences from France obtained from finding connected components formed an evolutionary lineage/clade, we first combined the 479 sequences sampled from France with 100 randomly-selected complete, high-coverage sequences sampled from outside France through March 17th, 2020 and uploaded to GISAID through April 29th, 2021. These sequences were aligned to Wuhan/Hu-1 using MAFFT, insertions were removed, and the sites described above were masked. This alignment was concatenated with the aligned sequences from France. IQ-Tree v. 2.0.7[62] was used to construct a maximum likelihood phylogeny, and ModelFinder[63] was used to find the best fit nucleotide substitution model (GTR + F + I). Small branches were collapsed. TreeTime v. 0.8.0[64] was used to remove any sequences with more than four interquartile distances from the expected evolutionary rate, rooting at Wuhan/Hu-1. Treetime was also used to generate a time-aligned phylogeny assuming a clock rate of $1 \times 10^{-3}$ substitutions per site per year with a standard deviation of $5 \times 10^{-4}$ substitutions per site per year, a skyline coalescent model, marginal time reconstruction, accounting for covariation, and resolving polytomies. Maximum likelihood phylogenies were visualized in Python using Matplotlib v. 3.3.3[65] and Baltic (https://github.com/evogytis/baltic).

**Reporting summary**
Further information on research design is available in the Nature Portfolio Reporting Summary linked to this article.

## Data availability
The simulated data generated in this study are available at https://github.com/koellelab/segregating-sites. The transmission pair data used to estimate the per-genome, per-transmission event mutation rate is provided in Table S1. The SARS-CoV-2 viral genome sequences used in the France analysis are available from GISAID (Supplementary information; https://doi.org/10.55876/gis8.230123mt). Due to the size of datasets, source data (excluding genome sequences downloaded from GISAID) are available at https://github.com/koellelab/segregating-sites.

## Code availability
Python code used for generation of all figures is available on GitHub: https://github.com/koellelab/segregating-sites.

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

## Acknowledgements
The research reported in this paper was supported by the National Institute of General Medical Sciences grant NIH/NIGMS R01 GM124280 R01 GM 12480 and R01 GM124280-03S1 (K.K.), the National Institute of Allergy and Infectious Diseases Centers of Excellence for Influenza Research and Response (CEIRR) contract # 75N93021C00017 (K.K.), and NIH NIAID F31AI154738 (M.A.M.). We thank the Koelle lab, Aaron King, Sally Otto, and Ailene MacPherson for feedback, as well as the BIRS Mathematics and Statistics of Genomic Epidemiology workshop for the opportunity to discuss this work.

## Author contributions
Y.P.: Conceptualization, Methodology, Software, Validation, Formal Analysis, Investigation, Writing, Visualization. M.A.M.: Conceptualization, Methodology, Software, Formal Analysis, Investigation, Writing, Visualization. K.K.: Conceptualization, Methodology, Investigation, Writing, Supervision.

## Competing interests
The authors declare no competing interests.
