## [Peer Review File · Nature Communications]

Epidemiological inference for emerging viruses using segregating sitesREVIEWER COMMENTS

Reviewer #1 (Remarks to the Author):

This is an interesting study that reports a novel method for estimating epidemiological parameters using sequence data, but in the absence of phylogenetic trees. In particular, the authors devise the trajectory of the number of segregating sites according to a model and a set of parameters. Using simulations, they demonstrate the accuracy of this approach and compare it to those based on phylodynamic inference (i.e. using trees).

I have no major concerns and no recommendations for additional analyses, but there are a few important points that I do believe the authors could develop further to improve the scope of this work, which is very promising!

1- In typical phylodynamic approaches using birth-death models, sequence sampling times are very informative, via the sampling rate. I suggest explaining the difference between such sampling rate and using sampling windows. The former seems more intuitive and realistic to me, but I might be missing something important that can be better explained here .

2- The estimation of the time of the index case, t_0 , is an interesting one, but I am not sure the comparison with the tmrca from phylodynamic methods is the most appropriate one. The tmrca corresponds to the first known transmission event, whereas t_0 is the time of the first infection, so closer to the origin parameter, which is the tmrca + length of the root branch. I am not about the accuracy of this parameter in practice using birth-death or coalescent models (usually not estimated in the latter), but it is an important consideration when discussing t_0 . Moreover, because the origin is usually estimated in stochastic models, it may be less sensitive to the push-of-the-past.

3- One thing that I kept thinking about while reading this is that in some viruses we see a lot of molecular rate variation early on, which is probably not accounted for by a Poisson model. Can the authors speculate on whether this would impact these inferences very much? In phylodynamics we can use a relaxed-clock for example, but I have rarely found that it has a substantial impact in epi estimates for viruses that are rapidly evolving.

4- Particle filters are still in their very early days in phylodynamics and phylogenetics, but I agree that they have many advantages over the typical MCMC approaches that we use. If possible, it would be extremely valuable to include an illustration of the particle filter approach here and of how the likelihood is calculated. If this is too much work, then I suggest citing accessible literature on particle filters (e.g. Bouchard-cote 2014 in 'Bayesian Phylogenetics'). I can imagine many readers glossing over this, but it is very important to improve the uptake of these methods.

Reviewer #2 (Remarks to the Author):

This paper describes a comparatively lightweight tree-free method for estimating the reproductive number and other epidemiological parameters from genome sequence data using a summary statistic. The method assumptions mean it is particularly suited to the early phase of exponentially growing epidemics and should only be applied to neutrally evolving sites. As the authors mention, low genetic diversity at the start of an outbreak leads to high phylogenetic uncertainty, which complicates inference using traditional phylodynamic models. In addition, when genetic diversity is low the genetic distance tree has many polytomies, which means that Bayesian phylodynamic inferences can be overly biased by the tree prior used. Furthermore, it is now apparent that shortly after spillover host-editing processes, such as APOBEC-mediated editing in the case of Monkeypox, may play a larger role in observed mutations than the usual polymerase errors. Since this method doesn't need to model the mutation process it is not affected by the process responsible for generating mutations, as long as the process is neutral.

I think the method is interesting and widely applicable to new emerging diseases and new variants. In particular I was surprised at how much information there was in the segregating sites trajectory, even when the epidemic trajectories appear very similar.

I note that the main limitation that makes the approach only applicable to the start of epidemics or shortly after an introduction of a variant into a new location is the infinite-sites assumption that assumes that each new mutation is unique has not been seen before (in the dataset) and won't occur again before the end of the analysis. This assumption could be relaxed, but it would make the simulation step far more costly (possibly prohibitively so), and the resulting segregating sites trajectory may be less informative.

Major comments

1. The authors mention that the model would be violated if there was strong selection (when mutations occur more often at specific sites). Richard Neher has a recent preprint showing exactly that for SARS-CoV-2 (<https://www.biorxiv.org/content/10.1101/2022.08.22.504731v1>). Furthermore, he shows that the rate of adaptive evolution steeply decreased over the first few months of the pandemic, which would also be a violation of the model used here. However, he found that the synonymous rate has stayed roughly the same over the course of the whole pandemic, and can thus be used as a neutral marker. Moreover, similar effects have been observed for other viruses (e.g. Ebola virus and influenza virus). This suggests to me that the segregating sites method should only consider synonymous mutations or mutations at third codon positions. Of course this would drastically decrease the amount of data (number of segregating sites) and may reduce the power of the method on the French SARS-CoV-2 dataset.

2. There is a slight disconnect between the simulation study and the real-data example. The authors state multiple times that the method is only applicable to the early phase of an epidemic, and this is also how it is applied in the real-data example, where only the initial phase of exponential growth is considered (Fig. 9). However, in the simulation study the epidemic is run over its whole course (from $I = 0$ to $I = 0$, Fig. 4). The limitation that the method is only applicable to the early phase stems from the infinite-sites assumption, and therefore the model is still applicable to the entire simulated epidemic trajectory, where the infinite-sites assumption holds by design. However, I wonder if the early-phase of the segregating sites trajectory alone is still informative enough to accurately estimate the epidemiological parameters? The real-data example indicates that it may be, but this isn't examined in the simulation study.

3. I think it's a bit unrealistic to assume the basal genotype outside France stays the same over the whole time period when a lot of segregating sites accumulate inside France. Could this be better justified?

4. From my reading it sounds as if ambiguous sites are not treated as a mutation ("an R nucleotide was assumed to match both an A and a G"). That would make the number of segregating sites used a lower bound. It's also possible to get an upper bound by treating all ambiguous sites as a mutation. Does this cause any significant change in the estimates? This is complicated a bit by the presence of sequencing errors, which would somewhat inflate the number of segregating sites (even though the authors do some filtering for errors it's very likely that some would inevitably still slip through).

5. The authors state that the computational complexity of the method is largely independent of the number of input sequences. What is the exact relationship between complexity and the number of sequences and also the number of cases?

Minor comments

1. The method described here has some similarities to a method developed by Emma Saulnier and

Samuel Alizon a few years ago

(<http://journals.plos.org/ploscompbiol/article?id=10.1371/journal.pcbi.1005416>), specifically simulating epidemics with mechanistic models and using summary statistics of the simulation result to fit model parameters. However, that method used many different summary statistics derived from simulated phylogenies, and used ABC to fit the model. Whereas the Saulnier et al. method is applicable to all scenarios (not just the start of epidemics), I think the approach described here is computationally much less intensive (no need to simulate trees), and even though it only considers one summary statistic, that statistic appears to have enough power to correctly infer the parameters of interest.

2. The assumption that patients are only sampled at recovery is a little unrealistic. In reality most patients are sampled (and sequenced) shortly after the onset of symptoms. The authors can assume (as in most birth-death models used in phylodynamics) that patients quarantine after sampling and effectively become uninfecious (so that the estimated infectious period is affected by NPIs), but it can also be argued that this is unrealistic for an airborne disease like SARS-CoV-2. The authors mention that this assumption can be relaxed. What is the effect of sampling at different points in the disease progression, or sampling from multiple compartments?

3. HIV and SARS-CoV-2 is the virus (virus is part of the abbreviation), but Ebola, dengue and influenza are the diseases. Use Ebola virus etc.

4. I don't think the BEAST1 folks will appreciate being snubbed out of a citation. BEAST1 and BEAST2 are now quite different packages, being developed concurrently by (almost entirely) non-overlapping sets of authors. Of the two, BEAST1 is more geared toward infectious disease phylodynamics, even though BEAST2 has the more complicated birth-death and structured coalescent models. For BEAST2 please use the more up-to-date BEAST 2.5 paper (Bouckaert et al, PLOS Comp Biol 2019), which also mentions the PhyDyn package used here. For BEAST1 the BEAST 1.10 paper (Suchard et al) should be used.

5. How were the 95% CI boundaries on the 2-dimensional plots calculated? Naïvely I would think it should be a convex hull or at least a single polygon, but in some plots (Fig. 7 and 8) it even has multiple unlinked components and some strange edge effects excluding pixels at the border. Furthermore, in some figure captions (notably the first such figures) the caption doesn't mention it at all.

6. Seroprevalence is not equal to just the number of recovered individuals and it is likely that many patients in the I (and perhaps even the E) compartment would already have seroconverted. Therefore, it's likely that seroprevalence would be higher to what is shown here, and closer to the results from the serosurvey. In addition, although the clade examined here was the majority clade circulating in France at the time, there were other infections not belonging to this clade, which would also contribute to a higher true seroprevalence.

7. Fig. 1: The legends of some panels overlap with the trajectories. Using transparent lines in the legend makes it difficult to see which colour is which.

8. Fig. 6: Panel C is missing a label.

9. Fig. S1 may work better if the same y-axis scale was used for all three panels.

RESPONSE TO REVIEWER COMMENTS

Reviewer #1 (Remarks to the Author):

This is an interesting study that reports a novel method for estimating epidemiological parameters using sequence data, but in the absence of phylogenetic trees. In particular, the authors devise the trajectory of the number of segregating sites according to a model and a set of parameters. Using simulations, the demonstrate the accuracy of this approach and compare it to those based on phylodynamic inference (i.e. using trees).

I have no major concerns and no recommendations for additional analyses, but there are a few important points that I do believe the authors could develop further to improve the scope of this work, which is very promising!

1- In typical phylodynamic approaches using birth-death models, sequence sampling times are very informative, via the sampling rate. I suggest explaining the difference between such sampling rate and using sampling windows. The former seems more intuitive and realistic to me, but I might be missing something important that can be better explained here.

We thank the reviewer for this comment. We agree that sequence sampling times are highly informative of model parameters and population sizes in birth-death models. In our segregating sites approach, sequence sampling times are not particularly informative in-and-of-themselves. This is largely because sampling of individuals in our model does not feed back to the underlying epidemiological dynamics. We assume that individuals are sampled upon recovery, once they are no longer infectious. If sampling is extremely dense, however, the number of samples does decrease the likelihoods of low R_0 models because simulated infection numbers in those models would not be able to reproduce observed sample numbers, resulting in particle weights of zero (see Methods). In practice, however, sampling effort is rarely so dense as to substantially inform model parameters under our approach.

Because the number of segregating sites in a sampled set of sequences increases with the number of sequences sampled, our approach does need to control for sampling effort and it does so in a statistically appropriate way within the particle filtering algorithm (see Methods).

That sampling effort does not impact parameter inference we think is actually a benefit of our approach. This is because sampling effort and testing rates can change dramatically over the course of an emerging pandemic as surveillance efforts expand. As such, there is a non-constant probability of an infection being sampled.

In response to this comment, we have added a paragraph to the Discussion section (around line 422), highlighting these considerations:

While our inference approach does adopt assumptions of phenotypic neutrality and infinite sites, it does not assume a constant sampling rate or a specific sampling process throughout the time period over which sequences are collected. As we have shown in Figure 1B, sampling effort does impact the segregating sites trajectory: the greater the sampling effort, the larger the number of segregating sites. For our inference approach to perform effectively, sampling effort therefore needs to be matched between the simulations and the empirical data. This matching of sampling effort is implemented in the particle filter. However, the

number of samples sequenced per time window is not particularly informative of model parameters (except in the case of extremely high sampling effort when certain low R_0 model parameterizations cannot appropriately evaluate the expected number of segregating sites in a time window because the number of sampled sequences exceed the number of simulated infections). The reason why the number of samples is not particularly informative of model parameters is because, under our approach, sampling of individuals does not impact the underlying epidemiological dynamics: individuals are sampled upon recovery, once they are no longer infectious. That the number of observed samples is not highly informative of model parameters we see as a benefit of our approach because sampling effort and testing rates can change dramatically over the course of an emerging pandemic or over the early period of an emerging viral lineage as surveillance efforts ramp up. In contrast, sampling times of sequences have been shown to be highly informative of model parameters in the case of birth-death models, with sampling process misspecification resulting in the possibility of arriving at biased parameter estimates (Volz and Frost 2014).

2- The estimation of the time of the index case, t_0 , is an interesting one, but I am not sure the comparison with the tmrca from phylodynamic methods is the most appropriate one. The tmrca corresponds to the first known transmission event, whereas t_0 is the time of the first infection, so closer to the origin parameter, which is the tmrca + length of the root branch. I am not about the accuracy of this parameter in practice using birth-death or coalescent models (usually not estimated in the latter), but it is an important consideration when discussing t_0 . Moreover, because the origin is usually estimated in stochastic models, it may be less sensitive to the push-of-the-past.

We agree that a direct comparison between the time of the index case t_0 and the time of the most recent common ancestor (tMRCA) is not an appropriate one to make, and we recognize with this comment that we likely did not articulate that sufficiently well in our original submission. We have expanded the text around lines 211-216 to more comprehensively explain the difference between t_0 and tMRCA and to indicate that approaches incorporating demographic stochasticity (such as our approach and birth-death models) effectively accommodate the push-of-the-past effect by conditioning on an epidemic taking off.

The text edits include the following around this part of the manuscript read:

Because PhyDyn infers epidemiological parameters using a tree-based method, the program does not estimate the time of the index case t_0 . Instead, it estimates the time of the most recent common ancestor (tMRCA) of the viral phylogeny. The credible interval of PhyDyn's tMRCA estimate spanned from -28.22 to 0.96 days post the true time of the index case ($t_0 = 0$). Times of a most recent common ancestor, however, are generally later (and never earlier) than the time of the index case. This is because some basal viral lineages likely go unsampled and the pruning of these unsampled basal lineages results in a tMRCA that can be considerably later than the time of the index case t_0 (Pekar et al. 2021). As such, interpretation of the PhyDyn results would almost certainly result in timing the index case t_0 as less than 0 (too early), given 0.96 days as the top end of the tMRCA credible interval. The early estimate of t_0 may be due to the "push-of-the-past" effect (Nee et al. 1994), which results from the assumption of deterministic dynamics in the inference process when the underlying population dynamics are stochastic (and conditioned on the persistence of a lineage). This "push-of-the-past" effect is usually reflected in an overestimate of the growth rate (or an overestimate in R_0) in coalescent-based inference approaches

that are applied to datasets with small population sizes during their exponential growth phase (Boskova et al. 2014). Here, because R_0 controls not only the rate of increase in the number of infected individuals at the start of the simulated pandemic but also the time at which the simulated pandemic starts to decline, the “push-of-the-past” effect may instead be reflected in a tMRCA estimate that occurs too early. Because our inference approach implements stochastic population dynamics, it appropriately accounts for the push-of-the-past effect, as do phylodynamic inference approaches that incorporate stochastic population dynamics (e.g., birth-death models).

3- One thing that I kept thinking about while reading this is that in some viruses we see a lot of molecular rate variation early on, which is probably not accounted for by a Poisson model. Can the authors speculate on whether this would impact these inferences very much? In phylodynamics we can use a relaxed-clock for example, but I have rarely found that it has a substantial impact in epi estimates for viruses that are rapidly evolving.

This is a very interesting comment. Our Poisson model of mutation does not actually prohibit deviations from a molecular clock. We model mutations as occurring at the point of transmission, with the mean number of mutations that arise (at the consensus level) being given by the parameter μ . We do not expect this mean number to change over the course of emerging pandemic (unless the viral polymerase evolves to have either higher or lower fidelity, which is not commonly observed in emerging pandemics). However, we agree that there can be changes in the molecular clock early on in a pandemic, where evolutionary rates are measured in terms of substitutions/site/unit time. These changes in the rate of molecular evolution are likely due to changes in the generation interval as individuals’ awareness of the pandemic increases or non-pharmaceutical interventions (NPIs) are implemented. If the generation interval (the time between infection and onward transmission) becomes shorter and mutations occur at transmission, this would lead to an increase in the rate of molecular evolution at the population-level. This is because there would be more transmissions per unit time. In contrast, if the generation interval lengthens, then the rate of molecular evolution at the population-level would decrease, because there would be fewer transmissions per unit time.

We have added text to the Discussion section to explain this in greater detail (around line 453):

Finally, we would like to note that setting the per genome, per transmission mutation rate to a constant value does not correspond to an assumption of a constant molecular clock. A constant molecular clock requires that the number of substitutions per unit time remains the same. Our assumption is that the mean number of nucleotide changes that occur during a transmission event between a donor and a recipient (at the consensus level) stays constant over time. This would almost certainly be the case unless the fidelity of the viral polymerase was evolving over the period considered. Changes in the substitution rate could come about if the generation interval between transmission events changes due, for example, to implementation of non-pharmaceutical interventions or increased symptom awareness. A shortening of the generation interval (defined as the time between infection and onward transmission) would increase the number of transmission events that occur per unit time and thereby result in an increase in the substitution rate. In contrast, a lengthening of the generation interval would result in fewer transmission events occurring per unit time, thereby decreasing the substitution rate. Changes in the generation interval can emerge from an underlying epidemiological model, such that our assumption of a constant per genome, per

transmission event mutation rate does not preclude or conflict with the observation of changes in the substitution rate over time.

4- Particle filters are still in their very early days in phylodynamics and phylogenetics, but I agree that they have many advantages over the typical MCMC approaches that we use. If possible, it would be extremely valuable to include an illustration of the particle filter approach here and of how the likelihood is calculated. If this is too much work, then I suggest citing accessible literature on particle filters (e.g. Bouchard-cote 2014 in 'Bayesian Phylogenetics'). I can imagine many readers glossing over this, but it is very important to improve the uptake of these methods.

We thank the reviewer for this comment and have added a supplemental figure (Figure S10) to the manuscript that graphically depicts the particle filter inference approach we use. We have also included an extensive figure legend with this supplemental figure to explain the steps implemented in our approach. We allude to this supplemental figure from the Methods section.

Reviewer #2 (Remarks to the Author):

This paper describes a comparatively lightweight tree-free method for estimating the reproductive number and other epidemiological parameters from genome sequence data using a summary statistic. The method assumptions mean it is particularly suited to the early phase of exponentially growing epidemics and should only be applied to neutrally evolving sites. As the authors mention, low genetic diversity at the start of an outbreak leads to high phylogenetic uncertainty, which complicates inference using traditional phylodynamic models. In addition, when genetic diversity is low the genetic distance tree has many polytomies, which means that Bayesian phylodynamic inferences can be overly biased by the tree prior used. Furthermore, it is now apparent that shortly after spillover host-editing processes, such as APOBEC-mediated editing in the case of Monkeypox, may play a larger role in observed mutations than the usual polymerase errors. Since this method doesn't need to model the mutation process it is not affected by the process responsible for generating mutations, as long as the process is neutral.

I think the method is interesting and widely applicable to new emerging diseases and new variants. In particular I was surprised at how much information there was in the segregating sites trajectory, even when the epidemic trajectories appear very similar.

I note that the main limitation that makes the approach only applicable to the start of epidemics or shortly after an introduction of a variant into a new location is the infinite-sites assumption that assumes that each new mutation is unique has not been seen before (in the dataset) and won't occur again before the end of the analysis. This assumption could be relaxed, but it would make the simulation step far more costly (possibly prohibitively so), and the resulting segregating sites trajectory may be less informative.

We thank the reviewer for this comment and agree with their assessment. We have added a couple of sentences to the Discussion section to highlight these points (starting at line 416):

We would also like to note that the infinite sites assumption could in principle be relaxed, but this would make the simulations in the inference approach substantially more costly. Furthermore, as time goes on, not only do chances of repeated mutations at sites increase, but genetic diversity increases. As such, phylogenetic uncertainty will decrease, such that existing tree-based phylodynamic inference approaches will become increasingly informative and segregating site trajectories less informative.

Major comments

1. The authors mention that the model would be violated if there was strong selection (when mutations occur more often at specific sites). Richard Neher has a recent preprint showing exactly that for SARS-CoV-2 (<https://www.biorxiv.org/content/10.1101/2022.08.22.504731v1>). Furthermore, he shows that the rate of adaptive evolution steeply decreased over the first few months of the pandemic, which would also be a violation of the model used here. However, he found that the synonymous rate has stayed roughly the same over the course of the whole pandemic, and can thus be used as a neutral marker. Moreover, similar effects have been observed for other viruses (e.g. Ebola virus and influenza virus). This suggests to me that the segregating sites method should only consider synonymous mutations or mutations at third codon positions. Of course this would drastically decrease the amount of data (number of segregating sites) and may reduce the power of the method on the French SARS-CoV-2 dataset.

We thank the reviewer for this comment. Our inference approach, like most phylodynamic approaches (including those relying on birth-death models and those relying on coalescent models), assumes that observed genetic variation is neutral. Not all genetic variation is of course neutral. The preprint the reviewer alludes to shows that, for SARS-CoV-2, there is direct evidence of purifying selection (less nonsynonymous variation is observed than would be expected under neutrality; Figure 4 of the Neher preprint). Neher also concludes that adaptive (positive) selection might have been occurring early on based on trends in the rate of nonsynonymous evolution over time. While it is true that Neher found that the synonymous rate of evolution stayed roughly constant over the course of the pandemic, we do not want to restrict ourselves to considering only third codon positions for a number of reasons. First, as the reviewer points out, this would decrease the amount of data substantially, and as a result the confidence intervals of parameter estimates would balloon. Second, we used a per genome, per transmission mutation rate of $\mu = 0.33$, which was parameterized based on observed transmission pairs. This mutation rate estimate considered all nucleotide changes in the SARS-CoV-2 viral genome, not just the subset of synonymous ones. As such, and because we would effectively be masking 2/3 of the genome, our per genome mutation rate would need to be recalculated and it would become much lower, thereby again posing difficulties for inference. Third, because there is genetic linkage between loci across the SARS-CoV-2 genome, we are not convinced that masking nonsynonymous sites is a full-proof or statistically appropriate approach, as it ignores linkage effects and background selection. However, we agree with the reviewer that not all genetic variation is neutral and have decided to expand on the downstream consequences of this assumption on our parameter estimates. The expanded text in the Discussion now reads (starting at line 386):

Our inference approach relies on several assumptions that are shared by existing phylodynamic inference methods. Most notably, it relies on an assumption that all mutations are phenotypically neutral. However, a recent analysis of SARS-CoV-2 sequences has shown evidence for purifying selection, even early on during the pandemic (Ghafari et al. 2022). Indeed, within the set of SARS-CoV-2 sequences from France, we observe 170 nonsynonymous mutations and 138 synonymous mutations (a ratio of 1.23:1). Given the number of nonsynonymous sites ($n = 68540$) and the number of synonymous sites ($n = 19255$) in the SARS-CoV-2 genome, we would expect, under neutrality, a ratio of 3.56:1. This underrepresentation of nonsynonymous genetic variation points towards purifying selection in our analyzed dataset. A more recent preprint also raises the possibility of adaptive evolution occurring during early 2020 (Neher 2022). Incorporating non-neutral genetic variation into inference approaches such as ours and existing phylodynamic ones is complicated, although some newer statistical approaches have started to tackle this goal (Rasmussen and Stadler 2019). In the context of our segregating sites inference approach, directly incorporating non-neutral evolution will increase model complexity considerably, and assumptions would need to be made about the distribution of mutational fitness effects. Rather than incorporating non-neutral evolution within our approach, we can for now consider how the occurrence of non-neutral evolution would impact our parameter estimates. With purifying selection at play, we would expect to see less genetic variation than if it were absent. As such, the number of segregating sites in any time window would be lower than it would be under neutrality. Our inference approach, assuming neutrality, would therefore bias R_0 estimates to be low and, in single-introduction models, the timing of the index case t_0 to be late. In multi-introduction models, our estimate of η would be biased high.

2. There is a slight disconnect between the simulation study and the real-data example. The authors state multiple times that the method is only applicable to the early phase of an epidemic, and this is also how it is applied in the real-data example, where only the initial phase of exponential growth is considered (Fig. 9). However, in the simulation study the epidemic is run over its whole course (from $I = 0$ to $I = 0$, Fig. 4). The limitation that the method is only applicable to the early phase stems from the infinite-sites assumption, and therefore the model is still applicable to the entire simulated epidemic trajectory, where the infinite-sites assumption holds by design. However, I wonder if the early-phase of the segregating sites trajectory alone is still informative enough to accurately estimate the epidemiological parameters? The real-data example indicates that it may be, but this isn't examined in the simulation study.

We thank the reviewer for this comment, which made us recognize that we did not sufficiently emphasize that we also applied the segregating sites approach to a much shorter simulated time series (an early subset of the time series shown in Figure 4). Our results on this shorter simulated time series was shown in Figure 5. We have now expanded Figure 5 to include an additional panel (labeled A). This panel shows the segregating sites trajectory over this shorter time period of sampling, with the remaining panels (B and C, previously A and B), showing that segregating sites inference still works on this short time series. Of note, while the confidence intervals on our parameter estimates for this very short dataset are (as expected) much wider than on the whole time series (Figure 5B (new) vs. Figure 3A), the true values still lie within the 95% confidence region. We further added a set of 3 more panels to this figure, showing results on a similarly short time series of segregating sites, with a higher mutation rate of $\mu = 0.4$. Inference based on this higher mutation rate resulted in a tighter 95% confidence region for the R_0 and t_0 parameter estimates, with the true values again falling into this region. The

simulations shown in Figure 5, on a short time series, with a μ of 0.2 and a μ of 0.4 bookend the mutation rate of $\mu = 0.33$ that we use in our SARS-CoV-2 application. They are also similar to the SARS-CoV-2 application in the number of time windows considered.

3. I think it's a bit unrealistic to assume the basal genotype outside France stays the same over the whole time period when a lot of segregating sites accumulate inside France. Could this be better justified?

We thank the reviewer for this comment. We have added text to the manuscript to better justify this assumption. The text is provided around line 738:

Our simplifying assumption that only the basal genotype can be introduced into France from the outside reservoir ignores the possibility that genotypes that are derived from the basal genotype enter France from the outside reservoir. Strictly speaking, we think this assumption is unlikely to be met. However, at very early time points in France's epidemic, most of the genotypes outside of France should still be the basal genotype, and only at later time points should the frequencies of derived genotypes increase outside of France. Introduction of these derived genotypes at these later timepoints could result in the establishment of viral sublineages in France. However, because autochthonous infections would be high at this point, these viral sublineages would very likely go unsampled. As such, we do not think that our assumption of only the basal genotype being introduced into France would have a dramatic effect on our results. We can consider, however, the effects that violation of this assumption would have on our parameter estimates. If derived genotypes were introduced into France and sampled (or their descendants sampled), then the number of segregating sites that would have evolved within France would be lower than we are currently taking it to be. As such, our current estimate of R_0 would be biased high.

4. From my reading it sounds as if ambiguous sites are not treated as a mutation ("an R nucleotide was assumed to match both an A and a G"). That would make the number of segregating sites used a lower bound. It's also possible to get an upper bound by treating all ambiguous sites as a mutation. Does this cause any significant change in the estimates? This is complicated a bit by the presence of sequencing errors, which would somewhat inflate the number of segregating sites (even though the authors do some filtering for errors it's very likely that some would inevitably still slip through).

We thank the reviewer for this comment. As a response to this comment, we considered alternative approaches for considering ambiguous nucleotides in the calculation of the number of segregating sites. If 'N's are treated as mutations, the number of segregating sites in each time window increases considerably (Figure S11A). However, based on our experience, it is very likely that an 'N' reflects insufficient read depth or quality to call a nucleotide, rather than a mutation. If we only consider the subset of non-N ambiguous nucleotides, and treat these as mutations, the segregating sites trajectory is barely affected (Figure S11B). As such, we do not believe that our parameter estimates using the France sequence data would be appreciably impacted by our current approach of treating ambiguous nucleotides as matching existing variation. We have added a Figure S11 to the manuscript to show these analyses, and refer to this supplement figure from the Methods section with the following text (starting at line 800):

Ambiguous nucleotides were considered in the calculation of segregating sites, with an ambiguous nucleotide assumed to match existing variation that is present when possible. An 'N' nucleotide was therefore assumed to match all four nucleotides, whereas an 'R' nucleotide was assumed to match only 'A' and 'G' nucleotides. This matching assumption results in a lower bound estimate for the number of segregating sites in any time window. If we instead count an 'N' nucleotide at a site as a mutation, the number of segregating sites in each time window is much larger (Figure S11A). However, it is unlikely that an 'N' nucleotide captures a mutation; it is much more likely that an 'N' indicates an inability to call a nucleotide based on low read depth or poor quality scores at a site. If we still count 'N' nucleotides as matching observed variation but count other ambiguous nucleotides such as 'R' as mutations, the segregating site trajectory is barely affected (Figure S11B). This is because there are very few non-'N' ambiguous nucleotides in the dataset. As such, our parameter estimates on the France dataset are very unlikely to be impacted by our assumption of ambiguous nucleotides matching observed genetic variation at their respective sites.

5. The authors state that the computational complexity of the method is largely independent of the number of input sequences. What is the exact relationship between complexity and the number of sequences and also the number of cases?

The computational complexity of the method is indeed largely independent of the number of input sequences: with a larger number of sampled sequences, over the same time period studied, the number of time windows would not change. During each time window, there would be k grabs of n_i sequences to calculate the mean number of segregating sites for the time window for a given particle. The number of sequences n_i would be higher in each time window i , but this increase is not computationally expensive. The computational complexity of the model instead depends largely on the number of genotypes present in the model: a larger number of genotypes leads to longer runtimes. The number of genotypes is larger for any given segregating sites trajectory if the particle filter is parameterized with a higher R_0 , with an earlier time of the index case t_0 , or with a higher contact rate factor η between the outside reservoir and a focal region. Computational complexity also increases with a higher value of the per genome, per transmission mutation rate μ . We have not determined an exact relationship between the number of genotypes and computational complexity, and do not think there is one that could be determined because it is not only the total number of genotypes simulated that matter but also when they arise in the simulation that would impact simulation speed. However, in response to this comment, we have added text to the discussion section that expands on the factors that impact the computational complexity of our method (starting at line 362):

One benefit of using such an approach for parameter estimation of emerging viral pathogens is that, early on in an epidemic, phylogenetic uncertainty present in time-resolved viral phylogenies is significant, and tree-based phylodynamic inference approaches would need to integrate over this uncertainty. This is often times computationally intensive, especially when many sequences have been sampled. The computational complexity of our "tree-free" approach, in contrast, does not scale with the number of sampled sequences. (Instead, the runtime required for parameter inference depends on the number of genotypes that evolve over the course of the model simulations. This number in turn is affected by the proposed basic reproduction number, the proposed time of the index case in the single introduction model, and the magnitude of the per genome, per transmission mutation rate μ .)

Minor comments

1. The method described here has some similarities to a method developed by Emma Saulnier and Samuel Alizon a few years ago (<http://journals.plos.org/ploscompbiol/article?id=10.1371/journal.pcbi.1005416>), specifically simulating epidemics with mechanistic models and using summary statistics of the simulation result to fit model parameters. However, that method used many different summary statistics derived from simulated phylogenies, and used ABC to fit the model. Whereas the Saulnier et al. method is applicable to all scenarios (not just the start of epidemics), I think the approach described here is computationally much less intensive (no need to simulate trees), and even though it only considers one summary statistic, that statistic appears to have enough power to correctly infer the parameters of interest.

We agree, and now also cite this paper when we mention other approaches that rely on population genetic summary statistics for sequence-based inference.

2. The assumption that patients are only sampled at recovery is a little unrealistic. In reality most patients are sampled (and sequenced) shortly after the onset of symptoms. The authors can assume (as in most birth-death models used in phylodynamics) that patients quarantine after sampling and effectively become uninfected (so that the estimated infectious period is affected by NPIs), but it can also be argued that this is unrealistic for an airborne disease like SARS-CoV-2. The authors mention that this assumption can be relaxed. What is the effect of sampling at different points in the disease progression, or sampling from multiple compartments?

We thank the reviewer for this comment. We agree that most sampling, at least during this early 2020 time period for SARS-CoV-2, likely occurred shortly after the onset of symptoms. However, the proportion of infections that were identified during this early period is thought to be extremely low (due to testing limitations as well as asymptomatic infections), such that sampling in-and-of-itself as a process very likely did not considerably impact the rate of viral spread, even if all 308 of the sampled patients quarantined after sampling. We agree with the reviewer, however, that our assumption of patients being sampled upon recovery may impact parameter estimates. We have expanded the text in the Discussion section to expand upon this point and to describe how sampling that occurred at other time points could have biased our parameter estimates (starting at line 442):

While the number of sampled sequences is largely uninformative of model parameters, our approach does have to make an assumption of when individuals are sampled. In our simulated dataset and in our application to SARS-CoV-2, we assumed that individuals were sampled as they recovered. This sampling scheme decision was based on our understanding that the time of symptom onset often follows peak viral load for many emerging viral pathogens (Linton et al., 2021) and an assumption that most testing early on in a pandemic involves individuals who develop symptoms. It is important to note that if the assumed sampled scheme is mismatched with the empirical sampling scheme, parameter estimates may be biased. For example, if individuals were instead sampled as they transitioned from the exposed class to the infectious class, rather than upon recovery, and we assumed in our model that individuals were sampled upon recovery, then our R_0 estimates would be biased high.

3. HIV and SARS-CoV-2 is the virus (virus is part of the abbreviation), but Ebola, dengue and influenza are the diseases. Use Ebola virus etc.

We thank the reviewer for this comment and have corrected this in the Introduction.

4. I don't think the BEAST1 folks will appreciate being snubbed out of a citation. BEAST1 and BEAST2 are now quite different packages, being developed concurrently by (almost entirely) non-overlapping sets of authors. Of the two, BEAST1 is more geared toward infectious disease phylodynamics, even though BEAST2 has the more complicated birth-death and structured coalescent models. For BEAST2 please use the more up-to-date BEAST 2.5 paper (Bouckaert et al, PLOS Comp Biol 2019), which also mentions the PhyDyn package used here. For BEAST1 the BEAST 1.10 paper (Suchard et al) should be used.

We thank the reviewer for this comment. We now also cite BEAST1 and have updated our reference for BEAST2.

5. How were the 95% CI boundaries on the 2-dimensional plots calculated? Naïvely I would think it should be a convex hull or at least a single polygon, but in some plots (Fig. 7 and 8) it even has multiple unlinked components and some strange edge effects excluding pixels at the border. Furthermore, in some figure captions (notably the first such figures) the caption doesn't mention it at all.

The 95% confidence intervals in the 1-dimensional plots are calculated using a chi-square distribution with 1 degree of freedom. The 95% confidence intervals in the 2-dimensional plots are calculated using a chi-square distribution with 2 degrees of freedom. We have added this to the figure legends. In the 2-D plots, although the set of cells that fall within this range would ideally be contiguous, because we take means of 20 log-likelihood values (a single value for each of 20 particle filter iterations for a given parameter set) and due to the underlying model dynamics being stochastic, each particle filter iteration will yield a slightly different log-likelihood value.

6. Seroprevalence is not equal to just the number of recovered individuals and it is likely that many patients in the I (and perhaps even the E) compartment would already have seroconverted. Therefore, it's likely that seroprevalence would be higher to what is shown here, and closer to the results from the serosurvey. In addition, although the clade examined here was the majority clade circulating in France at the time, there were other infections not belonging to this clade, which would also contribute to a higher true seroprevalence.

We thank the reviewer for this comment, and have added the following text to the Results section (starting at line 347):

Of note, our estimates should fall on the low side of these independent estimates because other, smaller clades were also circulating in France during the time period studied and infections with viruses from these other clades would also contribute to seropositivity levels. We also emphasize that this is

necessarily a rough comparison because seroconversion does not occur exactly at the point of recovery. It can occur over a broader range of times, ranging from prior to recovery to many days following symptom onset (Iyer et al. 2020).

7. Fig. 1: The legends of some panels overlap with the trajectories. Using transparent lines in the legend makes it difficult to see which colour is which.

We have edited this figure for clarity.

8. Fig. 6: Panel C is missing a label.

We have added a label for Panel C.

9. Fig. S1 may work better if the same y-axis scale was used for all three panels.

We have edited this figure such that all three panels have the same y-axis scale.

REVIEWERS' COMMENTS

Reviewer #1 (Remarks to the Author):

The authors have addressed multiple comments raised by both reviewers, including me. I am satisfied with their edits to the manuscript and would like to recommend it for publication.

However, after submitting my initial review I remembered an other approach to estimating epidemiological parameters with sequence data, but no trees. The method apparently was never published a peer review manuscript (Plazzotta and Colijin 2017; <https://www.biorxiv.org/content/10.1101/102061v1>). It is based on a correlation between the number of cherries in a tree and R_0 . Using sequence divergence, Plazzotta and Colijin estimate the number of cherries in a sequence alignment (if one had indeed estimated a tree) and then use this to infer R_0 . I suggest mentioning that method here for the sake of completeness and because tree-free methods are not hugely popular yet.

Other than that, I am very pleased about the new version of the manuscript and am confident that the methods developed here will make a valuable contribution to the field.

Reviewer #2 (Remarks to the Author):

The authors have addressed all of my comments and I'm happy with the manuscript. I don't have any additional comments, but I do have a few notes that I think would improve the manuscript.

1. From the main text I got the impression that when applying the segregating sites model and Phydyn to the early phase of the simulated epidemic (Fig. 5), that the segregating sites model performed better at estimating R_0 than Phydyn. Looking at the figure, both methods have similarly wide intervals for R_0 , and although Phydyn's estimate does appear to be more biased, both methods are biased to estimate higher R_0 values than the truth. This could perhaps be better explained in the main text. (Regarding t_0 the segregating sites methods clearly outperforms Phydyn).
2. Instead of speaking of a "simulated pandemic" I think it would be more appropriate to speak of a "simulated epidemic" here, since I think the dynamics in a pandemic are highly dependent on population structure and the movement of people between locations.
3. The preprint by Richard Neher has now been published: <https://academic.oup.com/ve/advance-article/doi/10.1093/ve/veac113/6887176>
4. There is a follow-up publication on the Rasmussen and Stadler (2019) paper that applies the model to SARS-CoV-2: <https://academic.oup.com/ve/article/7/2/veab073/6363035>
5. Line 75: time series -> time series
6. Line 241: Extra full stop after "rate"
7. Line 356: often times -> oftentimes
8. Line 434: sampled scheme -> sampling scheme
9. Line 600: particles -> particle
10. Line 620: from template -> from a template

RESPONSE TO REVIEWER COMMENTS

We thank the reviewers for their thorough reading of our work and for their additional comments on our revised manuscript. Below, we respond to these remaining comments. In addition to the edits we have made in response to these comments, we have identified two small inaccuracies in our code during the process of finalizing the manuscript. We have corrected these inaccuracies and their correction does not qualitatively impact the results of our analyses, although some quantitative results have been updated. The two inaccuracies were:

1. While generating simulated datasets, we sampled individuals with replacement, rather than without replacement. Our inference code, however, assumes that individuals are sampled without replacement. We corrected this inconsistency by sampling individuals without replacement in regenerated simulated datasets, and we re-ran all the analyses on these new simulated datasets. Figures 1-5 and Supplementary Figures 1-7 have thus been updated. As indicated above, correction of this inconsistency did not qualitatively change any of our results, although it did impact our quantitative estimates slightly.
2. In our empirical application to the sequence dataset from France, we had inadvertently been using a dataset that omitted several sequences. We thus updated the sequence dataset and re-ran our analyses based on this dataset. Figures 7-9 have thus been updated. While this updating did not qualitatively change any of our results, it did impact our quantitative estimates slightly.

Text in the Results section has been modified, where needed, in response to these changes.

We have also moved Figure 5d-f into the supplement (Figure S8) to ensure that the reader clearly recognizes that this is an additional analysis using time series from a previously unanalyzed simulation. In the main text, we further motivate and discuss this additional analysis in a slightly different manner than previously, as the update we made to this figure indicated that PhyDyn's performance improved with a higher mutation rate ($\mu = 0.4$).

Reviewer #1 (Remarks to the Author):

The authors have addressed multiple comments raised by both reviewers, including me. I am satisfied with their edits to the manuscript and would like to recommend it for publication.

However, after submitting my initial review I remembered another approach to estimating epidemiological parameters with sequence data, but no trees. The method apparently was never published a peer review manuscript (Plazzotta and Colijin 2017; <https://www.biorxiv.org/content/10.1101/102061v1>). It is based on a correlation between the number of cherries in a tree and R_0 . Using sequence divergence, Plazzotta and Colijin estimate the number of cherries in a sequence alignment (if one had indeed estimated a tree) and then use this to infer R_0 . I suggest mentioning that method here for the sake of completeness and because tree-free methods are not hugely popular yet.

We thank the reviewer for this comment. We now cite this preprint in the subsection entitled 'Brief overview of inference approach', along with citations of several other inference approaches that are based on summary statistics.

Other than that, I am very pleased about the new version of the manuscript and am confident that the methods developed here will make a valuable contribution to the field.

Reviewer #2 (Remarks to the Author):

The authors have addressed all of my comments and I'm happy with the manuscript. I don't have any additional comments, but I do have a few notes that I think would improve the manuscript.

1. From the main text I got the impression that when applying the segregating sites model and Phydyn to the early phase of the simulated epidemic (Fig. 5), that the segregating sites model performed better at estimating R_0 than Phydyn. Looking at the figure, both methods have similarly wide intervals for R_0 , and although Phydyn's estimate does appear to be more biased, both methods are biased to estimate higher R_0 values than the truth. This could perhaps be better explained in the main text. (Regarding t_0 the segregating sites methods clearly outperforms Phydyn).

We thank the reviewer for this comment. First, we have re-run the analyses for Figure 5a-c (please see our comments at the start of this response letter). Second, we now show Figures 5d-f as a supplemental figure (Figure S8). The text changes we have made in response to this comment and in response to the updated Figure S8 results are the following (starting on line 215 in the revised manuscript):

Because the impetus for developing the segregating sites inference approach was based on the extent of phylogenetic uncertainty present early on in an epidemic, we re-applied the inference approach to sequences sampled early on during the simulated epidemic, with time window bins ending on days 36, 40, 44, 48, and 52 (Figure 5a). During each of these five time windows, we sampled 10 sequences, resulting in a total of 50 sampled sequences. Our results on this subset of simulated data indicate that R_0 and t_0 could again be jointly estimated, although the confidence intervals for R_0 and t_0 were both considerably broader, as expected with a much shorter time series (Figure 5b). Similarly, on this same subset of data, Phydyn's 95% credible intervals were considerably broader (95% credible interval for $R_0 = 1.48$ to 10.80). For this particular time series, both the segregating sites approach and Phydyn tended to overestimate the true value of $R_0 = 1.6$ (Figure 5b, 5c). For Phydyn, the "push-of-the-past" effect³⁴ may have contributed to the overestimation of R_0 .

To determine whether there might be an upwards bias in the estimation of R_0 using the segregating sites approach, we simulated an additional short dataset under the same epidemiological model structure and model parameterization, with the exception of the mutation rate μ , which we increased from 0.2 to 0.4. To calculate the segregating sites trajectory, we sampled from this simulation as we did for Figure 5a-c, with 10 sequences sampled in each of the five time windows (Figure S8a). The maximum likelihood estimates of R_0 using our segregating sites approach did not overestimate the true R_0 of 1.6 in this dataset, although the time of the index case was again estimated to be slightly later than the true value of $t_0 = 0$ (Figure S8b). Compared to the results on the $\mu = 0.2$ short dataset (Figure 5b), the 95% confidence region spanned over a similar extent of parameter space. Phydyn also did not overestimate

R₀ on this $\mu = 0.4$ short dataset (Figure S8c). Moreover, its 95% credible interval was considerably smaller than on the $\mu = 0.2$ short dataset. This result makes sense: at higher mutation rates, phylogenetic uncertainty is reduced and tree-based inference approaches are expected to improve. In contrast, a low-dimensional summary statistic such as the number of segregating sites cannot take advantage of the higher-dimensional structure present in the sequence data.

2. Instead of speaking of a "simulated pandemic" I think it would be more appropriate to speak of a "simulated epidemic" here, since I think the dynamics in a pandemic are highly dependent on population structure and the movement of people between locations.

We thank the reviewer for this comment and have replaced "simulated pandemic" with "simulated epidemic" throughout the manuscript.

3. The preprint by Richard Neher has now been published: <https://academic.oup.com/ve/advance-article/doi/10.1093/ve/veac113/6887176>

We thank the reviewer for this comment and have updated this reference.

4. There is a follow-up publication on the Rasmussen and Stadler (2019) paper that applies the model to SARS-CoV-2: <https://academic.oup.com/ve/article/7/2/veab073/6363035>

We thank the reviewer for this comment. We do not cite this paper at this point, however, because the sentence as it stands refers to developed inference methods (rather than applications of these methods). If the reviewer feels strongly, we can edit this section in the discussion section accordingly.

5. Line 75: times series -> time series
6. Line 241: Extra full stop after "rate"
7. Line 356: often times -> oftentimes
8. Line 434: sampled scheme -> sampling scheme
9. Line 600: particles -> particle
10. Line 620: from template -> from a template

We thank the reviewer for pointing out these typos and have corrected them.